# A proteome-wide structural systems approach reveals insights into protein families of all human herpesviruses

Timothy K. Soh[1,2,3,4], Sofia Ognibene [1,2,3,4], Saskia Sanders [1,2,3,4], Robin Schäper [1,2,3,4], Benedikt B. Kaufer [5,6] & Jens B. Bosse [1,2,3,4] ✉

Structure predictions have become invaluable tools, but viral proteins are absent from the EMBL/DeepMind AlphaFold database. Here, we provide proteome-wide structure predictions for all nine human herpesviruses and analyze them in depth with explicit scoring thresholds. By clustering these predictions into structural similarity groups, we identified new families, such as the HCMV UL112-113 cluster, which is conserved in alpha- and beta-herpesviruses. A domain-level search found protein families consisting of subgroups with varying numbers of duplicated folds. Using large-scale structural similarity searches, we identified viral proteins with cellular folds, such as the HSV-1 US2 cluster possessing dihydrofolate reductase folds and the EBV BMRF2 cluster that might have emerged from cellular equilibrative nucleoside transporters. Our HerpesFolds database is available at https://www.herpesfolds.org/herpesfolds and displays all models and clusters through an interactive web interface. Here, we show that system-wide structure predictions can reveal homology between viral species and identify potential protein functions.

The Orthoherpesviridae family comprises alpha-, beta-, and gammaherpesviruses containing nine human herpesviruses (see Table 1 for virus abbreviations and their respective subfamilies). These are medically important human pathogens, including viruses such as human cytomegalovirus (HCMV), the leading infectious cause of congenital disorders in the USA. Due to the large number of encoded genes (between 70 and 181, depending on the species), experimental structural information for most viral proteins is not available, severely hampering their functional analysis. Moreover, herpesvirus gene and protein nomenclature are very inconsistent between species. Even worse, some proteins of the same species have multiple names based on their genome position, molecular weight, expression in the infected cell, or packaging into the virion. This situation makes it difficult even for experts to draw parallels between viruses. While some

corresponding orthologues between species can be deduced from the UniProtKB annotation, this database is incomplete and sometimes inconsistent. Finally, herpesviruses also have a penchant for duplicating genes, further complicating annotation.

With the development of machine learning-based prediction software[1,2], the accuracy of the structural information has increased so that it can often be directly used for building wet-lab testable hypotheses, making it an invaluable tool for molecular biologists[3]. DeepMind, in cooperation with EMBL-EBI, has published over 200 million protein structures to facilitate this process, covering most of UniProtKB in an open-access database (https://alphafold.ebi.ac.uk). While this is an extraordinary resource for many fields, it currently excludes viral proteins such as the human herpesvirus proteomes.

Herpesvirus phylogeny has classically been sequence-based, such as a recent study that classified the human herpesvirus proteomes into

[1]Hannover Medical School, Institute of Virology, Hanover, Germany. [2]Centre for Structural Systems Biology, Hamburg, Germany. [3]Cluster of Excellence RESIST (EXC 2155), Hannover Medical School, Hanover, Germany. [4]Leibniz Institute of Virology (LIV), Hamburg, Germany. [5]Institute of Virology, Freie Universität Berlin, Berlin, Germany. [6]Veterinary Centre for Resistance Research (TZR), Freie Universität Berlin, Berlin, Germany. ✉e-mail: jens.bosse@cssb-hamburg.de

**Table 1 | Viral proteins used for structure prediction**

| Subfamily | Virus | Virus abbreviation | Strain | Proteome ID or reference sequence | Number of proteins |
|---|---|---|---|---|---|
| Alpha | Herpes simplex virus 1 | HSV-1 | 17 | UP000009294 | 77 |
| Alpha | Herpes simplex virus 2 | HSV-2 | HG52 | UP000001874 | 74 |
| Alpha | Varicella-zoster virus | VZV | Dumas | UP000002602 | 70 |
| Beta | Human cytomegalovirus | HCMV | Merlin | UP000000938 | 181 |
| Beta | Human herpesvirus 6A | HHV-6A | U1102 | NC_001664 | 90 |
| Beta | Human herpesvirus 6B | HHV-6B | Z29 | NC_000898 | 97 |
| Beta | Human herpesvirus 7 | HHV-7 | RK | NC_001716 | 84 |
| Gamma | Epstein–Barr virus | EBV | B95-8 | UP000153037 | 87 |
| Gamma | Kaposi's sarcoma-associated herpesvirus | KSHV | GK18 | UP000000942 | 86 |

homology families[4]. However, the current revolution in structure prediction and rapid structural similarity search algorithms[5] has vastly expanded the pool of sequences that can be interrogated by structure-based phylogeny[6–8]. When applied to large proteins or large phylogenetic distances, e.g. across the *Duplodnaviria* realm, this approach clustered species more consistently than the sequence-based ones[9]. Moreover, a recent study using this approach showed that orthopoxviruses acquired cellular proteins, with some losing their original function[10]. Furthermore, structure prediction of the glycoproteins of different flaviviruses identified novel and common mechanisms in the fusion machinery[8].

Here, we predicted the structures of all nine human herpesvirus proteomes, totaling 844 individual proteins. We critically evaluated their accuracy with stringent quality scores and thresholds. We clustered their predicted structures into structurally similar groups and analyzed them using a structural systems virology approach. Many similarities suggest a common ancestor and thus a homologous relationship, whether orthologous, i.e. due to speciation, or paralogous, i.e. due to gene duplication. We found new members to previous groups identified by sequence, such as the US22 family of HCMV. We could also identify unexpected groups with distinct folds, such as the HCMV UL112-113 family harboring a unique beta-barrel domain that is conserved in seven of the nine human herpesviruses. A detailed domain-level similarity search identified several subclusters of proteins harboring different numbers of duplicated domains, deletions, and acquisitions. Using structural similarity searches against cellular proteins, we propose new functions for uncharacterized proteins. Examples include the HSV-1 and HSV-2 US2 proteins, which potentially have dihydrofolate reductase (DHFR) activity. We also found cases of exaptation, such as the BMRF2 family of transmembrane proteins that share structural similarity to nucleoside transporters but are likely inactive and now have functions in cell–cell spread.

In this work, we create a structural database that is accessible through an open-access and searchable web interface: https://www.herpesfolds.org/herpesfolds. It offers interactive displays of the predicted structures and structural clusters. HerpesFolds is a curated database that groups the protein predictions based on structural similarity while linking their established names and groups. It also serves as a reference that can translate the complicated herpesvirus nomenclature for the expert and non-expert alike and sheds light on the relationships of all nine human herpesvirus proteomes. This work will aid in developing wet-lab testable hypotheses and gain new insights into herpesvirus biology and pathology.

## Results

### Most herpesvirus proteins can be confidently predicted

To generate proteome-wide predictions of prototypic strains of all nine human herpesviruses (Table 1), we initially used LocalColabFold[11], an implementation of AlphaFold2 that mainly uses a 40–60× faster multiple sequence alignments (MSA) generator (Fig. 1A). For each

protein, five models were generated. For most proteins, the associated predicted local distance difference test (pLDDT) plots of all five models were very similar. This indicated that the models converged in all five runs to a similar solution, indicating that the best solution was found (Supplementary Fig. 1A). To categorize the models objectively, we established three thresholds based on the pLDDT consistency between models and the predicted template modeling (pTM) scores. See the "Methods" section for more details. Of the 844 proteins, 249 (30%) failed these quality scores (Fig. 1B and Supplementary Data 1 Sheet 1). We used standard AlphaFold[1] (Supplementary Data 1 Sheet 2) to improve these predictions using the slower but more thorough Jackhmmer MSA generator algorithm as well as LocalColabFold[11] with additional recycles (Supplementary Data 1 Sheet 3). This resulted in good predictions for 95 (38%) of the previously rejected models. Therefore, the final dataset consisted of 690 proteins (82% of 844 proteins) with consistent models as indicated by the pLDDT plot (Supplementary Data 1 Sheet 4). We compared the proteins that failed and passed the quality scores for each threshold criterion (Fig. 1C–E). Most proteins that did not pass had low pTM and a small percentage of sequence with a pLDDT > 0.70. This correlation suggested that proteins low in folded regions, as indicated by a pLDDT > 0.70, had lower prediction confidence. The pLDDT score can also predict whether a protein region is structured or intrinsically disordered. As shown before, a pLDDT > 0.70 indicates a well-modeled region with a Cα root mean square displacement likely <1.5 Å[12]. Conversely, pLDDT scores <0.50 likely indicate intrinsically disordered regions (IDR)[1,13,14]. Consequently, the AlphaFold database has been used to predict protein disorder, such as in MobiDB[15]. IDRs have recently been implicated in essential processes such as replication compartment formation of HCMV by liquid–liquid phase separation[16]. Therefore, we determined which well-predicted proteins likely contain disordered regions by quantifying the percentage of pLDDT values each model has <0.50 or >0.70. As shown in Supplementary Fig. 2A, 197 (23%) of the predicted proteins likely contain a disordered region (cut-off >44% of total sequence), and 640 (76%) of the proteins are likely structured (cut-off >33% of total sequence) (Supplementary Data 1 Sheet 5). This can be compared to the 1–8% disordered proteins in bacterial proteomes, 2–11% in archaea, 23–28% in eukaryotes, and 45% in humans[15,17,18]. Examples of N-terminal or C-terminal structured domains in a disordered protein are illustrated (Supplementary Fig. 2B). We also found a subset of 100 proteins that did not pass the pTM threshold but showed converging model pLDDT plots, indicating that the predictions came to a final solution. These proteins contained extensive sequence stretches with a pLDDT <0.50, which indicated that these proteins are well-predicted, disordered proteins. Finally, to indicate transmembrane domains as well as signal peptides, all predictions were further annotated with DeepTMHMM (Supplementary Data 1 Sheet 6).

To illustrate the accuracy of the general prediction workflow, we compared our predictions to experimental data that were released

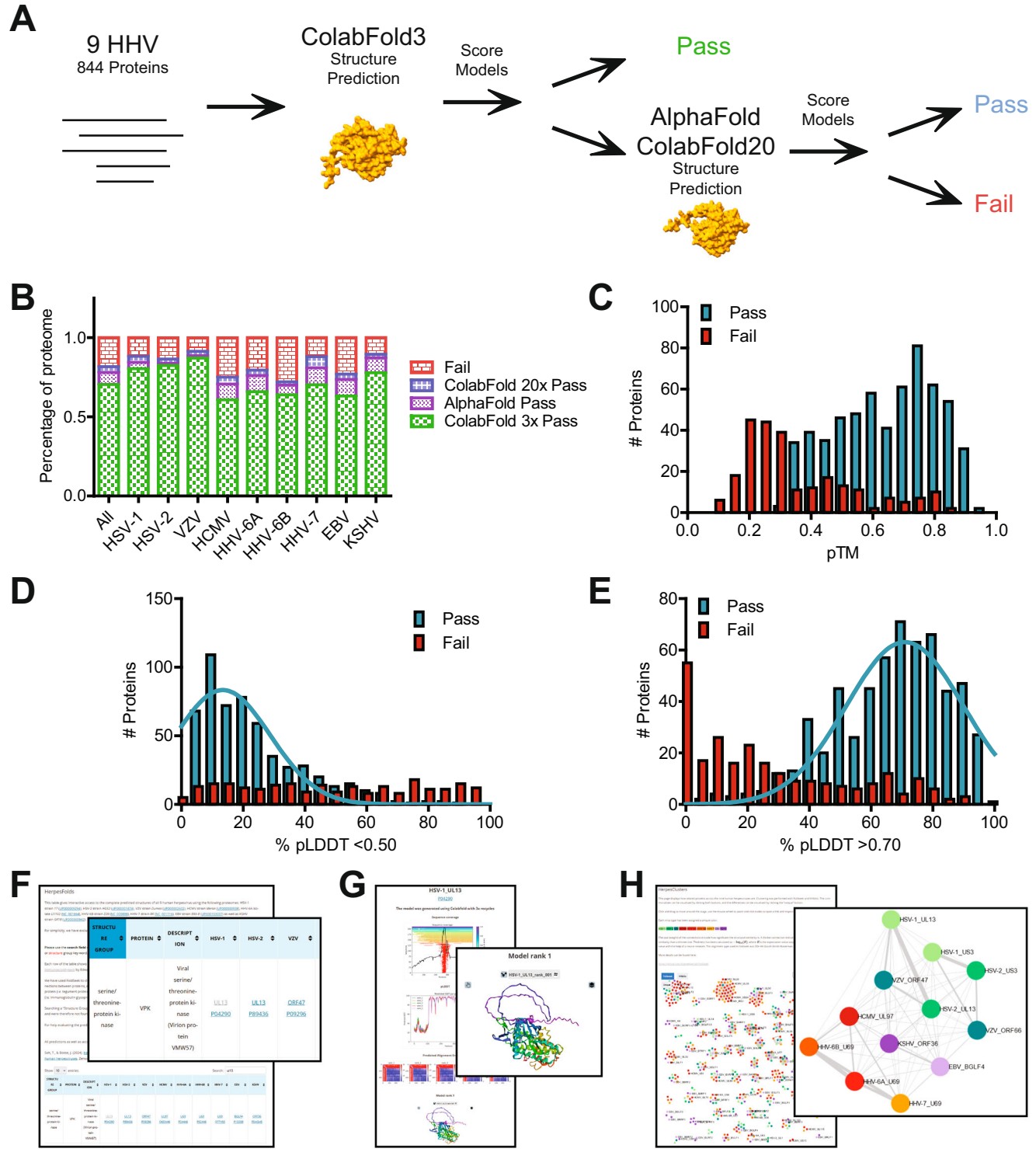

**Fig. 1 | Structure predictions of the proteome of all human herpesviruses.**
**A** Pipeline of model generation and quality scoring. The structures of 844 proteins covering the proteomes of all nine human herpesviruses were initially predicted with LocalColabFold with three recycles. After scoring, models that failed any quality score were rerun with AlphaFold and LocalColabFold with 20 recycles. **B** Percentage of each viral proteome that passed the quality score. **C** Histogram of the pTM scores of the initial LocalColabFold models that passed or failed the quality scores. Histogram of the percentage of each protein with a pLDDT score below 0.50 (**D**) or above 0.70 (**E**). In both cases, a Gaussian distribution was fit to identify the distribution for the proteins that passed the quality scores. **F** HerpesFolds online database is available at https://www.herpesfolds.org/herpesfolds. Structure predictions are organized and searchable by homology and structural similarity. **G** Example webpage of the displayed information for each predicted model. **H** Searchable clustering of all human herpesvirus proteins based on structure (Foldseek) or sequence (HHblits) similarity is available at https://www.herpesfolds.org/herpesclusters.

after the AlphaFold 2.3 training cut-off of 2021-09-30. Our predictions aligned very well with experimental structures, as depicted for HCMV UL77[19] (Supplementary Fig. 3). The Cα root mean square deviation (RMSD) (310 atoms) between the predicted and experimental HCMV UL77 structure (7NXP[19]) was 0.509 Å, indicating a highly consistent model. These observations are in line with a previous community assessment that concluded that AlphaFold models are, on average, as good as experimental structures[3].

## HerpesFolds is a searchable online structure database

To make our proteome-wide predictions accessible and easily searchable, we created a database at https://www.herpesfolds.org/herpesfolds (Fig. 1F). The main table contains the individual herpesvirus protein names and their UniProtKB accession numbers. Based on the previous knowledge[20,21], we generated a chart where each row contains proteins with known homology (Supplementary Data 2 Sheet 1). This way, the database also serves as a reference guide. This feature is important as the nomenclature of human herpesviruses is highly inconsistent and difficult to interpret even for experts. We also integrated the common names based on the UniProtKB annotation. Moreover, each protein entry links to the UniProtKB webpage and an individual webpage containing an animated model and the pLDDT and PAE graphs (Fig. 1G). If predictions were rerun, the best model is displayed, and if the models of all runs failed the thresholds, a warning is displayed at the top of the page. Furthermore, transmembrane domains as detected by DeepTMHMM are indicated and a second model lacking the signal peptide is provided. As an important feature, we also integrated a Foldseek[5] search link kindly provided by the Steinegger Lab that directly submits the predicted model to the Foldseek server and displays structurally similar proteins. Finally, we interactively display the structural similarity network (Fig. 1H).

## Clustering of known and unexpected structural homologs

The identification of herpesvirus orthologs has classically been through sequence alignment, but this approach is restricted by its need for some extent of conserved sequence similarity. To find more distant relationships that might represent homologs between different viral species and determine the potential function of orphan genes, we employed structural similarity searches. We used both DALI[22] as an established algorithm and the newer Foldseek[5] algorithm and ran all models against each other. We used a threshold of DALI Z score of >8 and Expect value (E value) < 0.001 for Foldseek (see "Methods" section) (Supplementary Data 2 Sheets 4 and 5, respectively). Overall, the resulting structural clusters were similar, with 615 proteins in protein similarity clusters for DALI and 632 for Foldseek. One example of a difference is the tegument protein 3 (TEG3) group. Foldseek grouped only the alphaherpesvirus TEG3 proteins with the gammaherpesvirus BGLF3.5/ORF35 proteins. In contrast, DALI also found a similarity of the roseolovirus (HHV-6A, HHV-6B, and HHV-7) U68 proteins with this group. Since Foldseek performed slightly better overall and is computationally much more efficient, we used it for the remainder of this work. The resulting structural similarity connections were used to cluster the herpesvirus proteins into groups of structural similarity (Fig. 2A). All clusters are available for interactive exploration at https://www.herpesfolds.org/herpesclusters.

Based on the UniProtKB[20] and previous work[4,21], 39 protein groups were expected to be found in all 9 species. A more recent sequence analysis based on domain architecture identified 23 proteins with a consistent domain structure in all 9 species and 169 homology groups[4]. Using structural similarity searches, we identified 25 groups with exactly 1 protein from each human herpesvirus and 89 homology groups. Many of these clusters matched these previous attempts to categorize herpesvirus proteins in groups based on experimental data and sequence analysis, confirming our approach. An example is the

homologs of the alphaherpesvirus UL25 proteins, which are essential capsid-associated proteins and part of the portal cap[19]. We also found clusters of proteins with known and verified enzymatic functions, such as the herpesvirus serine/threonine kinases, named after the prototypic HSV-1 US3 and UL13 proteins (Fig. 2B). As only the protein models that passed the quality scores were used, not all groups contain exactly one protein from each species. This was seen for the proliferating cell nuclear antigen (PCNA)-like protein, large tegument protein, and tegument protein 7 clusters. For example, HCMV UL44 is missing from the PCNA group because it was deemed a low-confidence model.

Surprisingly, the DNA polymerase catalytic subunit and DNA helicase/primase complex-associated protein were found to be structurally similar (Fig. 2C). While these proteins have different biological functions, this clustering can be explained by partial sequence similarity in the C-terminus[23,24]. Furthermore, they are both similar to nucleases, with HSV-1 UL8 being homologous to the exonuclease domain of B-family DNA polymerases[24] and HSV-1 UL30 having RNase H activity[25]. Since all viruses have both of these genes, this suggests that while these genes likely share a common ancestor, they have diverged in sequence and function.

The small capsomere-interacting protein (SCP) is conserved in all herpesviruses. However, they clustered as distinct groups based on the subfamily (Fig. 2D). Genetic drift may have led to structurally distinct features, as the last common ancestor likely predates the divergence of the subfamilies. This was seen with both the sequence- and structure-based clustering.

A particularly interesting cluster of so far-unannotated proteins consisted of the likely homologs of HCMV UL112-113. HCMV UL112-113 is essential for viral replication by mediating replication compartment formation through phase separation at viral genomes, thereby recruiting the viral polymerase[16]. The poorly characterized roseolovirus U79 proteins and the alphaherpesvirus UL4/ORF56 proteins cluster with HCMV UL112-113. They shared an undescribed and conserved predicted beta-barrel (Fig. 3A). These herpesviruses also form replication compartments and have a similar subcellular localization[16,26], and their potential HCMV UL112-113 homologs would be prime candidates for further investigation.

Another cluster contained the gammaherpesvirus KSHV ORF73, also known as LANA, and EBV EBNA1 proteins that harbor a known DNA binding domain[27,28] (Fig. 3B). Our structure predictions indicate that this DNA binding fold is conserved amongst all human herpesviruses, and we further analyzed these proteins by generating AlphaFold 3 predictions of dimers[29] with DNA (Supplementary Fig. 4). The alphaherpesvirus UL3/ORF58 proteins showed similar surface charge distributions and DNA binding sites as KSHV and EBV. In contrast, the betaherpesvirus UL122 and U86 proteins, which have known IE2 transcriptional regulation activity[30,31], bound to DNA with a different orientation in the predictions. Surprisingly, the betaherpesvirus proteins UL117 and U84 were not predicted to bind DNA, which is consistent with the less prominent positive surface charge. Perhaps these proteins have lost the DNA binding activity and acquired a different function.

The HCMV phosphoprotein UL25 formed a group with its roseolovirus U14 homologs and clustered with the HCMV UL35 proteins, which likely arose via gene duplication, constituting the pp85 superfamily[32]. Strikingly, we also found HCMV UL32 and its roseolovirus U11 homologs to be structurally similar. UL32, also known as pp150, is a protein conserved in the betaherpesviruses. It binds directly to capsids as an inner tegument protein while bridging the full virion tegument via its disordered C-terminus[33]. It will be essential to dissect if the structurally similar UL25 and UL35 can also bind capsids and potentially regulate pp150 abundance at pentons, potentially confusing the reconstruction of virus capsid cryo-EM maps. Finally, the poorly studied KSHV ORF48 and likely the EBV BBRF2 protein, which was excluded from the clustering due to low prediction scores, also

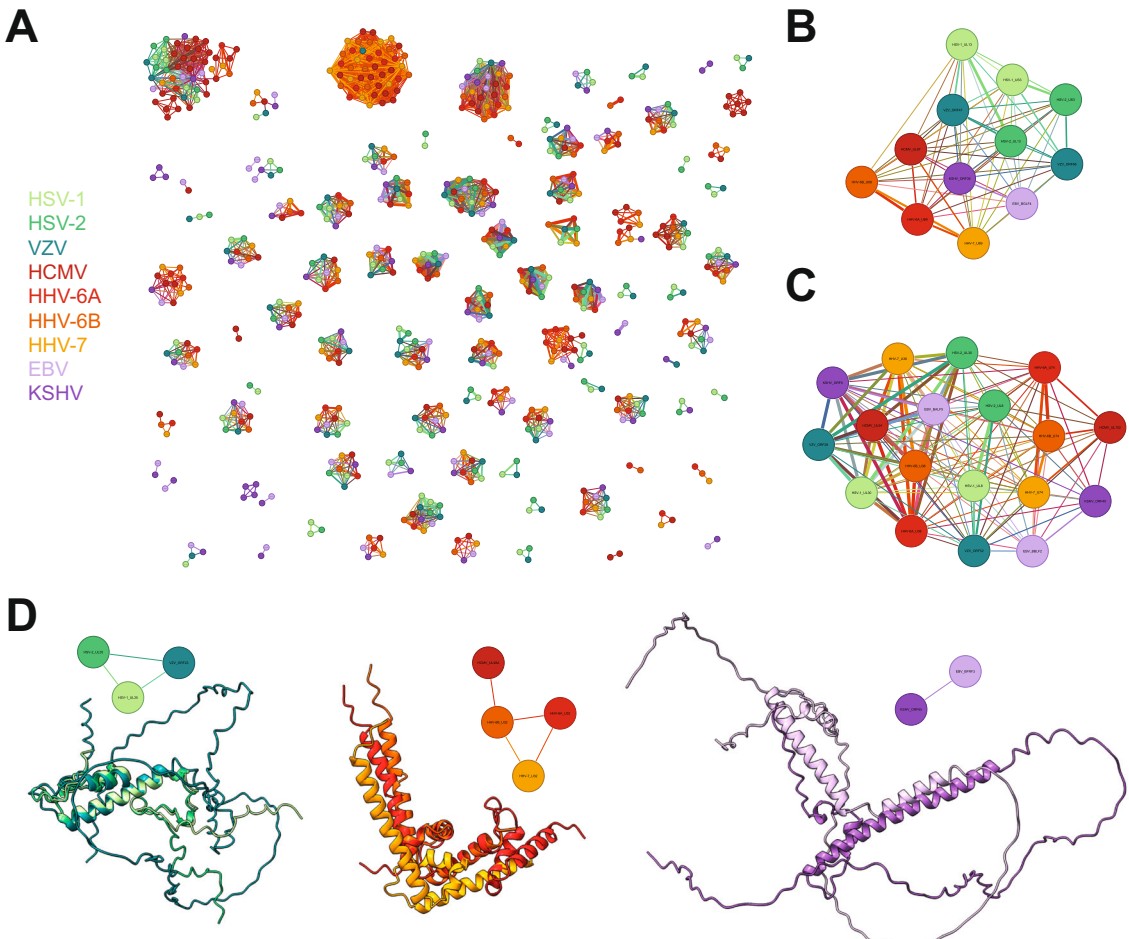

**Fig. 2 | Structural similarity search identifies protein clusters. A** Graphical display of all structurally similar protein clusters. The 690 herpes proteins that passed the quality score were analyzed pairwise by Foldseek for structural similarity. The color scheme for the viral species is used throughout this work. The name of the protein is written in the center of each node and is legible upon zooming in https://www.herpesfolds.org/herpesclusters allows interactive browsing of the clusters.

**B** Cluster of viral serine/threonine-protein kinases US3 and UL13. **C** Cluster of viral DNA helicase/primase complex-associated protein (HEPA) and DNA polymerase catalytic subunit (DPOL). **D** Clusters of small capsomere-interacting protein (SCP). The alpha-, beta-, and gammaherpesviruses clustered separately. The corresponding UCSF ChimeraX session can be found at https://zenodo.org/records/13284140.

clustered with the pp150 homologs. ORF48 is part of virions[34], and it should be assessed if it can bind to the viral capsid as pp150 does.

Since some protein predictions have unstructured regions, it is possible that these flexible regions lead to false negative results by Foldseek. To evaluate this possibility, all predicted structures were cut into smaller pieces based on where they are likely structured, i.e. pLDDT > 0.7. Foldseek clustering was then repeated and compared to clustering based on full-length predictions (Supplementary Data 2 Sheet 9). While this method resulted in an interesting novel cluster containing EBV BALF1, EBV BHRF1, and KSHV ORF16, which are all known apoptosis regulators, most novel clusters identified using this method were likely false-positives such as HSV-1 UL49 and UL49A. UL49 is a tegument protein involved in immune evasion, and UL49A is a glycoprotein, respectively. Another example is a cluster containing HCMV UL96 and HSV-1 UL14, which show little unique or similar tertiary structure. For this reason, we focused on the analysis of full-length proteins.

We also compared our results to sequence-based clustering by MMseqs2 and HHblits (Supplementary Data 2 Sheets 2 and 3, respectively). Importantly, several structure-based clusters contained additional proteins when compared to sequence-based clustering methods, such as the HCMV UL112-113 cluster and the KSHV ORF73 (LANA) cluster, where all the identified proteins of this cluster have functions related to DNA binding. However, we also identified cases

where structure predictions failed the quality scores and HHblits-derived clusters were more complete. For example, all pAP proteins, which are splice variants of the scaffold protein, failed the structure prediction quality scores and are absent from the structural cluster. Since sequence-based methods can complement these instances where the structure prediction is of low quality, we provide an option to view both the structure and sequence-based clusters at https://www.herpesfolds.org/herpesclusters. Supplementary Data 2 Sheet 10 lists the respective HHblits and Foldseek scores for each protein pair as well as all interactions above their respective thresholds, which were found only by HHblits, only Foldseek, or both.

**Immunoglobulin-like domains are common structural elements**
One of the largest clusters we identified using structural clustering via Foldseek consists of viral proteins that contain immunoglobulin (Ig)-like domains. This cluster merged several previously defined protein families. One of the included groups is the HCMV RL11 family[35]. They share an RL11 domain (RL11D), formed by a conserved tryptophan and two cysteines, and are structurally similar to immunoglobulin domains. The RL11 proteins are non-essential in cell culture[36] but have high sequence variability between isolates with some playing a role in modulating immune responses, such as UL11[37]. While there are 14 accepted members of this family, they did not form an isolated structural group in our clustering (Fig. 3C). Unexpectedly, EBV BILF2

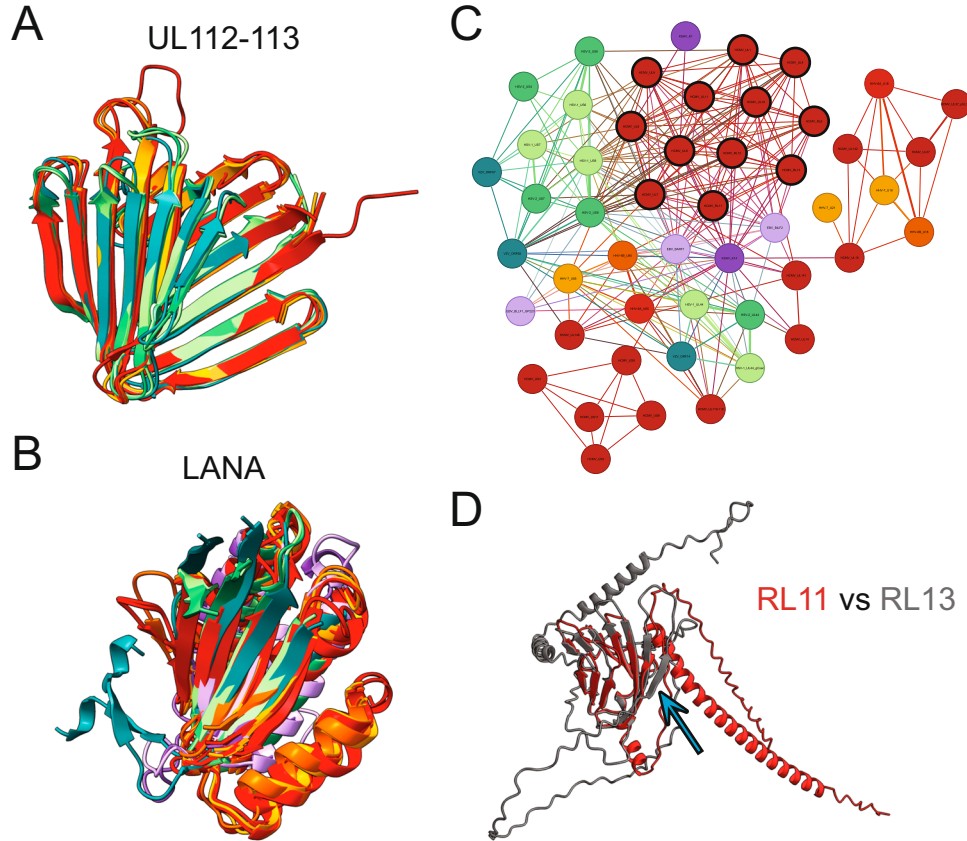

**Fig. 3 | Structural clustering identifies differently grouped protein groups.**
**A** Structural alignment of the HCMV UL112-113 cluster. Only the conserved beta-barrel-like domain is shown for clarity of HCMV UL112-113, HHV-6A U79_U80, HHV-6B U79, HHV-7 U79, HSV-1 UL4, HSV-2 UL4, and VZV ORF56. **B** Structural alignment of the LANA DNA binding domain-containing cluster with only the matching domain shown for clarity of KSHV ORF73, EBV EBNA1, HHV-6A U84, HCMV UL117, HCMV UL122, HCMV UL123, HHV-6A U86, HHV-6A U90_U87U86, HHV-6B U84,

HHV-6B U90_U86, HHV-7 U84, HHV-7 U86, HSV-1 UL3, HSV-2 UL3, and VZV ORF58. **C** Ig-like domain-containing cluster harboring the HCMV RL11 protein family. The canonical members are marked with a black outline. **D** Structural alignment of HCMV RL11 with HCMV RL13. The blue arrow points to the additional beta strand. The corresponding UCSF ChimeraX sessions can be found at https://zenodo.org/records/13284140.

and KSHV K14 but also the well-studied HSV-1 glycoproteins UL44 (gC), US6 (gD), US7 (gI), and US8 (gE) were structurally similar to multiple members of this family. EBV BILF2 is encoded in a genome region unique to *lymphocryptoviruses*, and KSHV K14 is also unique to KSHV. Furthermore, HSV-1 and HSV-2 US8 are located in a different part of the genome than HCMV. In addition to the low sequence identity, these data suggest that these Ig-like domain-containing proteins were acquired independently rather than from a common ancestor. Furthermore, gene duplication continued after speciation. While most HCMV RL11 members have the same 6-strand antiparallel beta-fold, HCMV RL13 has an additional beta strand (Fig. 3D), potentially representing a distinct evolutionary branch and function.

**Identification of domain-level duplications and deletions**
We performed the initial structural similarity search with full-length proteins. While this will identify commonalities between proteins, it will, by definition, not identify differences. To address how the domain architecture of a protein can differ between species, we performed a structural similarity search with overlapping structural snippets using a sliding window approach (see "Methods" section) (Supplementary Data 2 Sheet 6). We identified 2806 structurally similar connections between proteins, which includes 183 new connections for proteins relative to the full-length analysis (Supplementary Data 3 Sheets 1 and 2). This analysis identified 24 internal duplications, where domains were likely duplicated in the same protein; 35 repetitive acquisitions, where domains of a different protein matched the query protein

multiple times at different positions; and 13 domain additions, where multiple domains of a protein matched different proteins (Fig. 4A and Supplementary Data 3 Sheet 3).

We found that HSV-1 UL15 was structurally similar to the previously identified HSV-2 UL15 but also HSV-1 UL9 (Fig. 4B). UL15 is part of the tripartite terminase, essential for packaging viral genomes into newly formed capsids, and UL9 is the replication origin-binding protein. The fragment that aligns is oriented in the opposite direction relative to the core in UL15 versus UL9, which suggests that this region is not involved in an external interaction and has a conserved structural integrity function.

An important example of domain duplications is in the large betaherpesvirus US22 family cluster. No experimental structure of this large family of proteins is available, and sequence-based methods indicate that the family consists of at least 12 members in HCMV[38,39]. Using full-length structural similarity searches, we identified HCMV proteins that share the same conserved fold (Fig. 4C). Importantly, this group consists of four different domain architectures depending on whether the protein contains 0, 1, 2, or 4 copies of the US22 domain (Fig. 4D). The majority of proteins are similar to US22 with two domains, while the following members have only one domain: HCMV UL26, HCMV UL26_UL26.21, HCMV UL27, HCMV UL38, HHV-6A U19, HHV-6B U19, and HHV 7 U19. We also found a subgroup that lacks a US22 homology domain consisting of HHV-6A U4, HHV-6B U4, and HHV-7 U4. They are connected to the main group through the HHV-6A U7 and HHV-7 U7 proteins, which appear to have an additional domain

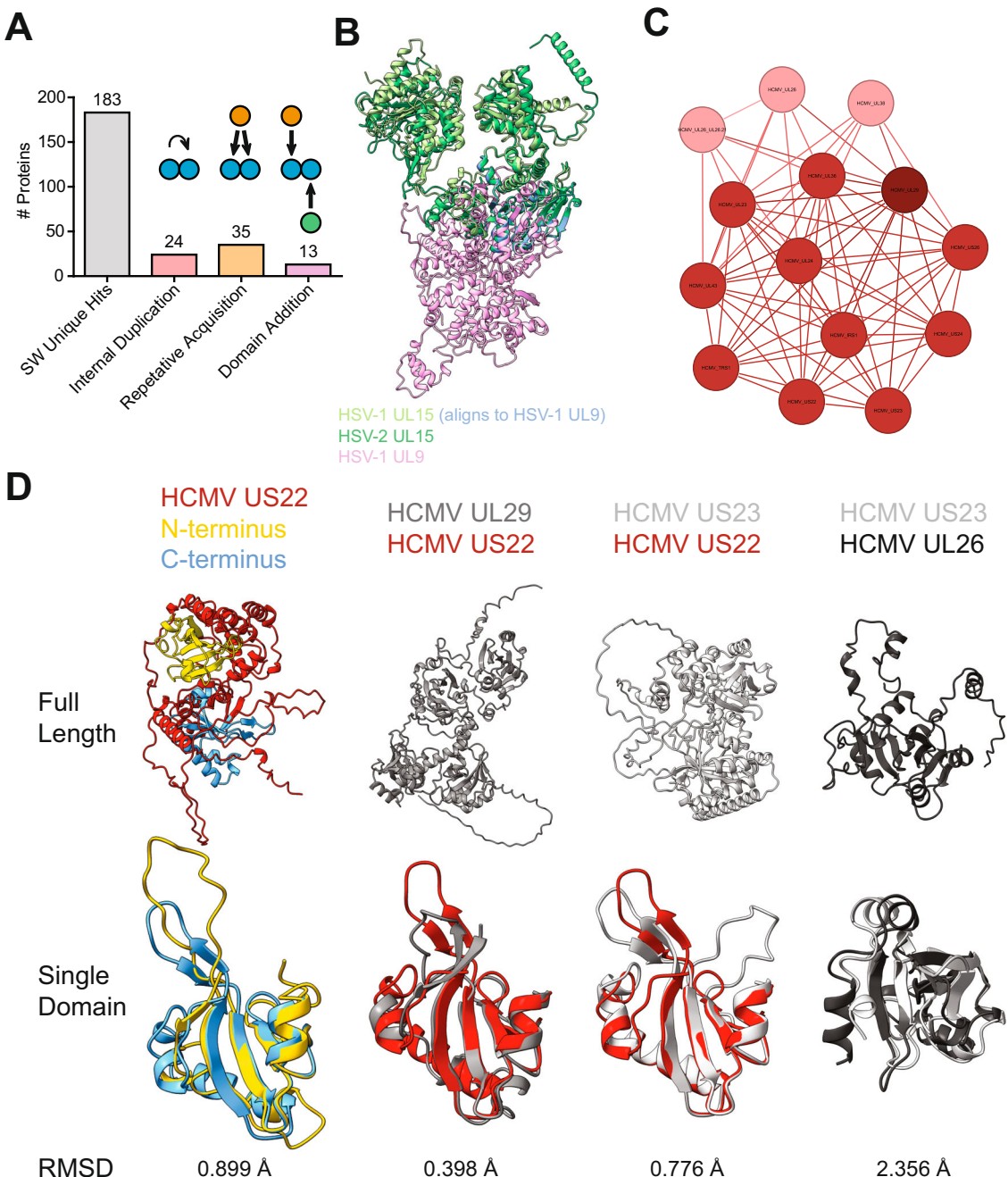

**Fig. 4 | Domain-level similarity search identifies conserved domain duplications. A** Domain-level similarities identified using a sliding window approach. The "SW Unique Hits" are the number of proteins that had a structurally similar query-target pair in the sliding window analysis that was absent from the full-length analysis. "Internal Duplications" are proteins where parts of the protein are similar to another part of itself. "Repetitive Acquisition" are proteins where a piece of a different protein matches the query protein multiple times at different positions. "Domain Addition" refers to proteins with multiple domains matching different proteins. Illustrative cartoons are shown above the corresponding bar. **B** HSV-1 UL15 is aligned to the known homolog HSV-2 UL15 and the structurally similar target HSV-1 UL9, which was only identified in the domain-level search. The part of HSV-1 UL15 that aligns with HSV-1 UL9 is colored blue. **C** The cluster of HCMV

proteins that contain HCMV US22. The proteins that contain 2 domains, like the canonical US22, are shown in the HCMV color. The proteins with one domain are shown in light red, and the protein that has four domains is shown in dark red. **D** The tertiary structure of the core domain is similar between the proteins in this group. The top row is the full-length protein, the bottom row is a structural alignment of the core domain, and the RMSD against the 91 Cα of the core domain alignment is written below. HCMV US22 has two domains, shown in yellow and blue, which are aligned. HCMV UL29 has four domains and was aligned to US22. HCMV US22 was aligned to HCMV US23, which has two domains, and HCMV US23 was aligned to HCMV UL26, which has one domain. In the case of multiple domains, the best-fitting domain combination is shown. The corresponding UCSF ChimeraX sessions can be found at https://zenodo.org/records/13284140.

relative to US22, which the U4 proteins consist of exclusively. The proteins with four copies of the US22 domain are HCMV UL29, HHV-6A U7, and HHV-7 U7. Oddly, HHV-6B U7, unlike the other roseoloviruses, only has two domains, suggesting a divergence in this species. Furthermore, many group members were identified to contain an internal

duplication and/or repetitive acquisition. The low RMSDs between the two domains of US22 and between the domains from different proteins are consistent with a domain duplication (Fig. 4D). These data suggest that the US22 family is bigger than previously thought and that domain additions and deletions are particularly common in this family.

## Viral exaptation of cellular proteins

Many of the identified clusters contained understudied proteins. We expanded the structural similarity search to cellular proteins using Foldseek to determine their potential biological activity from the PDB database (Supplementary Data 4 Sheet 1), and the predicted AlphaFold structures of the Swiss-Prot database (Supplementary Data 4 Sheet 3). The PDB contains experimentally verified structures, but it is limited in diversity. In contrast, the AlphaFold predictions of the Swiss-Prot database enable comparisons to a much more extensive predicted database that is less biased by experimental limitations. We combined the significant hits from the two databases and identified structurally similar cellular proteins. To find putative functions for the viral protein groups, the functional description of the homologous proteins was mined for keywords (Supplementary Data 4 Sheets 2 and 4). Unfortunately, the depositors' descriptions of structures in the PDB database are not systematic. To handle these variations, we calculated the frequencies of all words in the descriptions, and the words with the highest frequencies, which likely represent the protein function, were manually chosen as the annotation. Each viral homology group identified above was then given a HerpesFolds annotation. Since the PDB, but not the AlphaFold database, contains structures of viral proteins, it is important to note that some hits in the PDB database are because they were aligning with themselves.

Some predicted viral protein structures did not map to any other protein fold in the searched databases, indicating that these are unique folds (Supplementary Data 4 Sheet 5). We found that even though the viral alkaline nucleases are not similar to cellular nucleases, they are consistent within the herpesviruses (Fig. 5A). Experimental structures have been solved for EBV[40] and KSHV[41,42]. While the central core was found to show some similarity to λ-exonuclease, the viral proteins have additional terminal extensions. Another striking example was the cytoplasmic envelopment protein 2, which is conserved in all herpesviruses and for which no experimental structure exists (Fig. 5A). Still, this protein group was very well-predicted as indicated by its pLDDT and PAE scores. Moreover, the most distant phylogenetic pair, VZV ORF44 and HHV-6B U65, had an RMSD of 4.3 Å against the 112 Cα atoms despite a sequence identity of only 14%. These results indicate that AlphaFold can predict herpesvirus proteins with high confidence de novo even if no structurally similar dataset exists. Since the CEP2 proteins are crucially involved in virion envelopment[43], it will be important to analyze their role in modulating the cellular secretory system through this unique fold.

Using this workflow, we could annotate several poorly studied viral proteins and identify potential functions. One example is HSV-1 US2, which is structurally similar to DHFRs (Fig. 5B–D). The most significant cellular target was the DHFR from *Mesocricetus auratus* (commonly known as the golden or Syrian hamster) (PDB 3EIG[44]). Furthermore, HSV-1 US2 was structurally similar to KSHV ORF2, a known DHFR homolog with enzymatic activity[45,46]. While KSHV ORF2 contains the catalytic residues identified in *Escherichia coli (E. coli)*[47,48], HSV-1 and HSV-2 US2 do not contain them (Supplementary Fig. 5). Since no DHFR activity has been reported for the US2 proteins, this fold was likely exaptated. Functional assays will be needed to verify if the US2 proteins still have DHFR activity.

We also identified a cluster containing EBV BMRF2, KSHV ORF58, VZV ORF15, and HSV-1 UL43 (with HSV-2 UL43 being absent due to low-quality scores). Importantly, EBV BMRF2, KSHV ORF58, and VZV ORF15 predictions were significantly similar to the human equilibrative nucleoside transporter (ENT) 4 (Fig. 5E–G). This cellular protein is involved in transporting metabolites and regulating adenosine concentrations[49]. However, no experimental data currently supports a role for any of these viral proteins in transmembrane transport. Instead, EBV BMRF2 can change cellular morphology by modulating the actin cytoskeleton[50] and is essential for efficient cell–cell spread[51]. In a closely related alphaherpesvirus, UL43 colocalizes with proteins involved in membrane fusion and potentially interacts with the gM/gN complex[52], which is engaged in syncytia formation[53]. These commonalities suggest a function for this viral protein cluster in cell–cell spread, which does not align with the transmembrane transporter activity of the structurally similar cellular proteins. Moreover, comparative cross-sections of the viral proteins and ENT4 illustrate that they do not have a channel necessary for metabolite transport[54] (Supplementary Fig. 6A). Several transporters depend on accessory proteins for correct trafficking to the plasma membrane, such as the lysosomal transporter MFSD1, which depends on GLMP[55]. This is also true for EBV BMRF2, which needs BDLF2[50] for correct trafficking to the plasma membrane, and KSHV ORF58, which needs ORF27[34]. Interestingly, AlphaFold predictions of the viral heterodimers suggest that even if channels were present in BMRF2 and ORF58, they would be blocked by the partner protein (Supplementary Fig. 6B), arguing against a conserved transporter activity as recently proposed[56]. Instead, the structure predictions likely indicate that the similarity to ENT4 may stem from "radical" exaptation, where the transporter activity was lost and replaced by a role in viral cell-to-cell spread.

All human herpesviruses encode for deoxyuridine triphosphatases (dUTPases)[57] called HSV-1 UL50, HSV-2 UL50, VZV ORF8, HCMV UL72, U45 of roseoloviruses, KSHV ORF54 and EBV BLLF3. Cellular dUTPases are homotrimers that form three active centers at their interfaces. In contrast, the herpesvirus dUTPases are monomers consisting of a fusion of two dUTPase domains, as shown for EBV BLLF3[58]. While the respective alpha- and gammaherpesvirus proteins still have enzymatic activity, the betaherpesvirus homologs have lost them and are involved in immune modulation[59]. Human beta- and gammaherpesviruses code for 17 more dUTPase-related proteins (DURPs), likely due to gene duplications[57]. Using domain-level search, we found that all DURPs were predicted to harbor a three-domain architecture in contrast to the two domains of the viral dUTPases. However, the separate domains diverged quite significantly from the canonical dUTPase fold as indicated by their RMSD (Fig. 5H), which is in line with a recent report on the crystal structure of the HCMV DURP UL82 (pp71), which describes that it has lost its dUTPase function[60].

Herpesviruses have been suggested to induce autoimmunity through molecular mimicry[61,62]. Several studies have attempted to identify viral proteins that might induce autoimmunity due to a similarity to a cellular protein[63,64]. Protein sequence-based analyses could not identify viral proteins that would induce autoimmunity to known cellular proteins[63], such as myelin for multiple sclerosis[64]. Our structural similarity search identified no direct structural match of myelin to a viral protein. Interestingly, multiple members of the immunoglobulin domain-containing group, such as HSV-1 US8, mapped to myelin protein zero-like protein 1, and others, such as EBV BARF1, mapped to myelin-associated glycoprotein. These myelin-related proteins could be involved and warrant further investigation.

## Discussion

Understanding protein structures is critical to comprehending pathogen–host interactions. Unfortunately, experimental structural data are lacking for most human herpesvirus proteins. Machine learning-based structure prediction tools have made huge strides in accuracy and provide quality scores for their predictions[1,2,11]. Accordingly, recent community evaluations have found "that AF2 [AlphaFold2]-predicted structures, on average, tend to give results that are as good as those derived from experimental structures"[3] while also cautioning that "AlphaFold predictions are valuable hypotheses and accelerate but do not replace experimental structure determination"[12].

The human herpesviruses encode for at least 844 proteins, and for most, no experimental structure has been solved. Here, we applied proteome-wide predictions to provide a structural systems view of all annotated human herpesvirus proteins and developed stringent

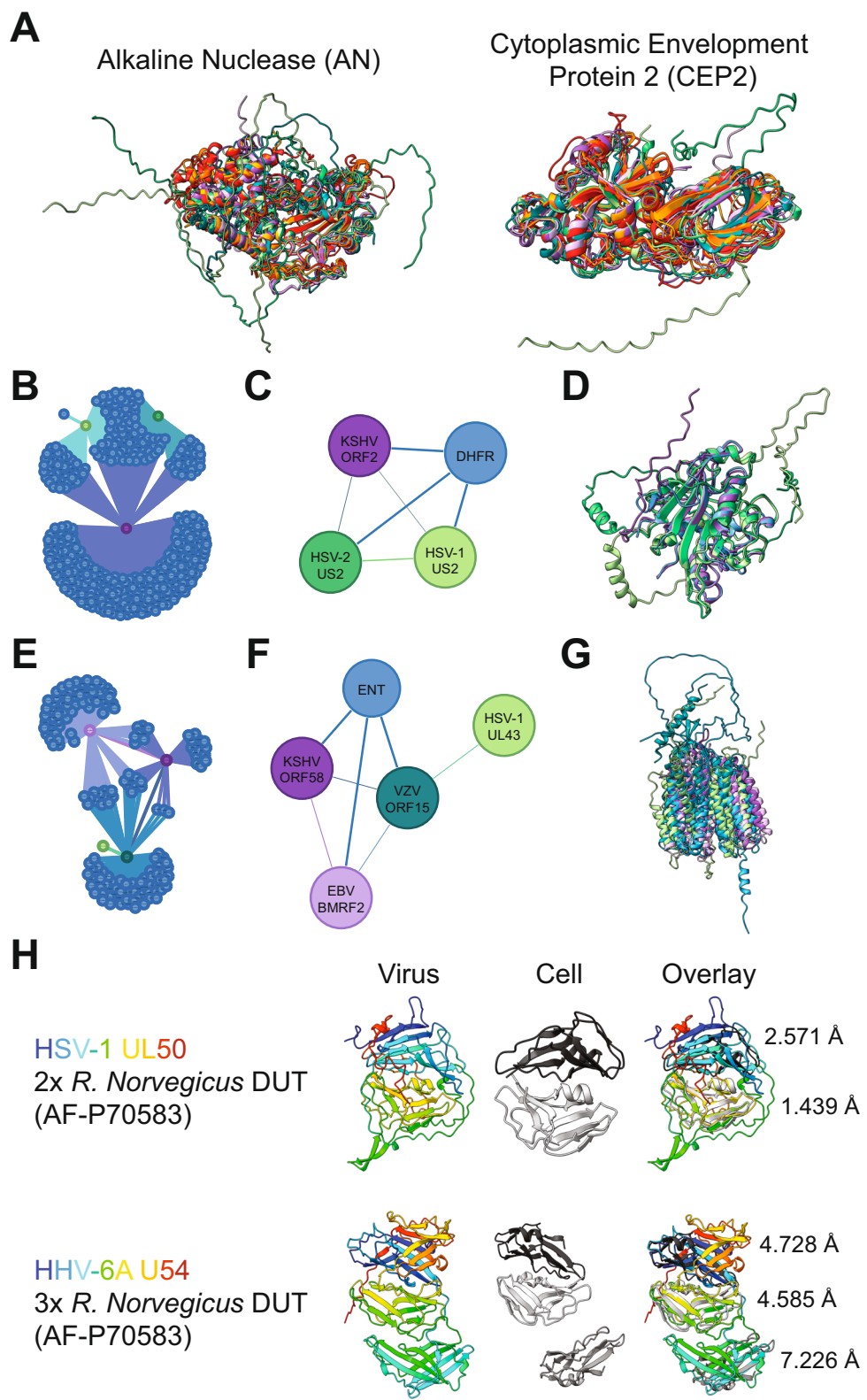

scoring workflows to validate our predictions. We generated clusters of proteins with similar folds using all-versus-all full-length structural similarity searches. Importantly, we identified previously unknown protein clusters sharing conserved folds, such as the HCMV UL112-113 family that carries a beta-barrel-like fold and likely plays an essential role in genome replication, while confirming previously known clusters, such as viral G-protein coupled receptors and kinases. In comparison, previous work has used sequence-based approaches to cluster herpesvirus proteins into related families. Importantly, neither a sequence-based alignment algorithm (MMseqs2) nor an HMM-HMM-based algorithm (HHblits) could uncover all of the novel and biologically important relationships that we found using a structural approach, such as the finding that the alphaherpesvirus UL3/ORF58 proteins are new members of the KSHV ORF73 (LANA) cluster, which are DNA-binding proteins and have too little sequence conservation to be identified by MMseqs2 or HHblits.

**Fig. 5 | Annotation of viral protein functions from structural similarity search.**
**A** Structural alignment of the alkaline nuclease (AN) and cytoplasmic envelopment protein 2 (CEP2) from all human herpesviruses. CEP2 was found in all human herpesviruses but had no significant cellular hit. **B** Cluster of KSHV ORF2, HSV-1 US2, and HSV-2 US2 containing the cellular hits. Each cellular node (blue) represents a specific protein. **C** Summary of (**B**) for clarity. **D** Structural alignment of the proteins in (**C**). P04753 dihydrofolate reductase (DHFR) was the most significant cellular hit. **E** Cluster of EBV BMRF2, KSHV ORF58, HSV-1 UL43, and VZV ORF15 containing the cellular hits. Each cellular node (blue) represents a specific protein. **F** Summary of

(**E**) for clarity. **G** Structural alignment of the proteins in (**F**). The most significant cellular hit was Q8R139 equilibrative nucleoside transporter 4 (ENT4). **H** Structural architecture of the classical DUT HSV-1 UL50 and the DURP HHV-6A U54. The rainbow-colored viral protein illustrates how the polypeptide folds back on itself. The most significant cellular hit, AF-P70583 DUT from *Rattus norvegicus*, is shown multiple times so that it can be aligned to each domain in the viral protein. The RMSD against the 205 Cα for each alignment is shown on the right of the overlay. The disordered termini were hidden for clarity. The corresponding UCSF ChimeraX sessions can be found at https://zenodo.org/records/13284140.

Still, structural clustering might not always indicate direct evolutionary relatedness. The large Ig-like domain cluster appears to be a group of proteins with a common fold, likely the result of independent acquisitions rather than originating from a single common ancestor or gene duplication. Using domain-level searches, we identified differences between structurally related proteins such as viral dUTPases and the HCMV US22 family. Both consist of subgroups with different numbers of duplicated domains that correlate with their described activity. Finally, we searched for cellular proteins with matching folds. We identified cases of potential exaptation, such as the membrane transporter ENT4 by human alpha- and gammaherpesviruses and unique folds that do not match any cellular protein, such as the CEP2 proteins.

This body of data provides many valuable hypotheses for experimental validation. To accelerate the use of this data and research on human herpesvirus, we have generated an easy-to-use, searchable web interface for the community, accessible at https://www.herpesfolds.org/herpesfolds. The database groups proteins by structural relationship and allows interactive evaluation of all structures. PDB files of each prediction can be downloaded or directly submitted to the Foldseek structural similarity search server to find similar cellular proteins in up-to-date databases.

This work is a first step based on high-confidence monomeric predictions of viral proteins. It neglects that some proteins may fold differently in complex with others, adopt multiple conformations, or need to form oligomers to function, such as gB[65]. Future work will need to take into account protein complex composition and stoichiometry. However, accomplishing this task is challenging as this information is unavailable for most viral proteins and generating all likely and possible combinations is computationally highly demanding. The next logical step will be predicting viral protein–protein interaction networks. Since post-translational modifications and protein–ligand complexes[66,67] or nucleic acids[68] will likely play a role in many such complexes, the recently released RoseTTAFold-All-Atom[69] and Alpha-Fold 3[70] algorithms will be of great use.

Herpesviruses have large bicistronic genomes, and for example, ribosomal profiling has identified 100s of additional transcripts and ORFs for HSV-1[71], HCMV[72], EBV[73], and KSHV[74,75]. Structure predictions of these could provide insights into their functions and roles during infection and pathogenesis and should be the next step.

The methodology presented here applies to any virus or virus family. It can easily be extended to analyze all reading frames of a given organism, not only the annotated ORFs. As the databases of predicted and experimental viral protein structures grow, more connections between different viral pathogens and their hosts will be made. The speed at which these in silico models can be generated will drive rapid hypothesis design and subsequent experimental confirmation.

## Methods
### Viruses
The strains and associated reference numbers of the nine human herpesvirus proteomes used for prediction are noted in Table 1. In addition to the full-length proteins, splice variants were also included. A complete list of the predicted proteins can be found in Supplementary Data 1 Sheet 4.

### Software
Sequence alignments were performed with MMseqs2 release 15-6f452[76] (https://github.com/soedinglab/MMseqs2) and HHblits v3.3.0[77] (https://github.com/soedinglab/hh-suite). HHblits HMM generation used the UniRef30 database from 2023_02. The significance threshold used for MMseqs2 was *E* value < 0.1[76] and for HHblits was *E* value < 0.001 (https://toolkit.tuebingen.mpg.de/tools/hhblits). Protein structure predictions were made with LocalColabFold v1.5.1[11] (https://github.com/YoshitakaMo/localcolabfold) and AlphaFold[1] using AlphaFold version 2.3.0 (https://github.com/google-deepmind/alphafold). Fasta files were downloaded from https://www.uniprot.org, and structure predictions were performed in accordance with the respective documentation detailed in the respective GitHub releases. For each protein, five models were generated using three recycles and a stop-at-score of 100. Proteins that failed the quality scores were rerun with LocalColabFold with 20 recycles. AlphaFold 3[70] predictions were performed through the AlphaFold Server at https://alphafoldserver.com. Protein topology was identified with DeepTMHMM[78] using their provided Demo Colab (https://dtu.biolib.com/DeepTMHMM). The custom Python script "DeepTMHMM_export_domains.py" consolidated the DeepTMHMM outputs into a single file and "DeepTMHMM_make_truncated_fasta.py" generated fasta files of the proteins without the identified signal peptide. Structural similarity searches were performed with local install versions of DALI[22] (DaliLite.v5 from http://ekhidna2.biocenter.helsinki.fi/dali/) and Foldseek[5] version 2ad017897d3dab66dd33ea675e92215bdfb4a64d (https://github.com/steineggerlab/foldseek). Networks were visualized with Gephi v0.10.1 (http://gephi.org/). UCSF ChimeraX version 1.8 (https://www.cgl.ucsf.edu/chimerax) was used for visualization of structures, and PyMOL 2.5.8 (https://www.pymol.org) was used for RMSD calculations. Data analysis was carried out with custom-written Python scripts, which were partly generated with the assistance of large language models and thoroughly tested manually. The complete set of scripts can be accessed through the project's GitHub repository: https://github.com/QuantitativeVirology/Herpesfolds.

### Evaluation of model quality
All proteins were initially predicted using LocalColabFold[11] with three recycles as it is more time-efficient. To assess the quality of the resulting models, three thresholds were used. Failing any single threshold led to the model being flagged as "fail". We interrogated whether the model optimization had converged by testing the consistency of the pLDDT scores associated with every model (Supplementary Fig. 1A, B). To do so, we first calculated the mean and standard deviation of the pLDDT values per model and then the standard deviation of these five values, resulting in "StDev of mean" and "StDev of StDev". To set thresholds, we fit a Gaussian curve to their respective histograms and set the cut-off at the mean of the Gaussian plus 2 times its standard deviation (Supplementary Fig. 1C, D). Therefore, a "StDev of mean" >3.2 or a "StDev of StDev" >1.9 was deemed a low-quality model. We also evaluated the pTM score to validate model confidence further. To set a global threshold at which a prediction likely constitutes a folded protein, we chose 106 herpesvirus proteins representing the protein length distribution of the herpesvirus proteomes and scrambled their amino acid

sequences. These scrambled sequences were used for model prediction and treated as a negative dataset, assuming that a random residue order should not fold. Receiver operating characteristic (ROC) analysis yielded an area under the curve of 91.3%, and the Youden index was used to set a threshold (Supplementary Fig. 1E). Following this analysis, a pTM < 0.3150 was deemed a low-quality model. We reanalyzed all models that failed the initial threshold in AlphaFold[1] and LocalColabFold[11] with 20 recycles and rescored them. For example, Supplementary Fig. 1F shows an improved model run in AlphaFold with higher pLDDT scores compared to Supplementary Fig. 1B. The corresponding scripts can be found in our HerpesFolds GitHub repository in the file "score_colabfold.py" at https://github.com/QuantitativeVirology/Herpesfolds.

### Defining disordered and structured content of models

The pLDDT value at a given residue indicates whether a region is predicted to be disordered (pLDDT < 0.50) or structured (pLDDT > 0.70). For the models that passed all quality scores, we generated histograms of the percentage of the protein below or above these values and fit a Gaussian to both curves. Since the majority of proteins are expected to be structured, a threshold, >44%, for proteins containing a disordered region was set at the mean of the distribution + 2× StDev for the percentage of protein sequence below 0.50, and a threshold, >33%, for proteins with a structured region was set at the mean − 2× StDev for the percentage of protein sequence above a pLDDT of 0.70. The script for extracting pLDDT values from the.json files associated with every prediction can be found in our HerpesFolds GitHub repository in the file "score_colabfold.py" at https://github.com/QuantitativeVirology/Herpesfolds.

### Extracting pLDDT domains from predicted structures

The structured domains from a predicted structure were extracted based on the pLDDT scores. The regions with a pLDDT > 0.7 were identified. If adjacent regions were connected by a disordered region, i.e. pLDDT < 0.7, that was shorter than 100 residues, the adjacent regions along with the disordered region were counted as a single domain. The corresponding script can be found as "extract_pLDDT_domains.py" in our HerpesFolds GitHub repository at https://github.com/QuantitativeVirology/Herpesfolds.

### Structural similarity search with DALI and Foldseek

Foldseek aligns protein structures with decreased computation time compared to alternative algorithms[5]. It provides different alignment algorithms, 3Di + AA Gotoh–Smith–Waterman and Tmalign, and measures of significance, such as the Expect value (E value) and probability of homology (prob). We tested both alignment algorithms with different thresholds and compared the protein clusters, Supplementary Data 2 Sheet 7. The Tmalign results did not agree with the sequence-based clustering and known literature. The E value and prob thresholds with 3Di + AA Gotoh–Smith–Waterman gave similar results, and thus we chose to continue with an E value threshold of 0.001 to be consistent with the literature[10], Supplementary Data 2 Sheet 8.

To cluster viral proteins into groups of structural similarity, we employed DALI as an established algorithm and Foldseek as a recent algorithm designed to work with vast predicted datasets with decreased computation times. With DALI, we used the recommended threshold of Z score > 8 for probable homologs[79], and with Foldseek, we used the previously published threshold of <0.001[10]. Since the results from both algorithms were similar, we only used Foldseek for virus-host and domain-level searches as DALI is orders of magnitude slower, which would have made these analyses computationally too expensive otherwise. Similarity searches against cellular proteins were done using the PDB database (Supplementary Data 4 Sheet 1), and the

predicted AlphaFold structures of the Swiss-Prot database (Supplementary Data 4 Sheet 3).

### Clustering of structure predictions by similarity

The Foldseek and DALI outputs are a list of query and target proteins with the associated significance value. We used a threshold of E value < 0.001 for Foldseek and a Z score > 8 for DALI to determine the significant results. Both reciprocal comparisons for each protein pair needed to be significant for inclusion in the clustering, i.e. A matches B and B matches A. If protein A was structurally similar to B and B similar to C, then A, B, and C were clustered. Once all clusters were identified, we converted the protein names to their row position in Supplementary Data 2 Sheet 1. This system allowed us to quickly determine whether a structural cluster contained proteins expected to group, i.e. all row numbers are the same, or not, i.e. the row numbers are different, based on UniProtKB annotations[20] and previously known homologies[21].

The associated script can be found in our HerpesFolds GitHub repository in the file "analyze_foldseek_for_structural_similarity_cluster.py" at https://github.com/QuantitativeVirology/Herpesfolds.

### Analysis of sliding window structural similarity

To search for alterations at the domain level between protein predictions, we generated "structural snippets" from all predictions using a sliding window approach. We then ran these structural snippets in an all-versus-all Foldseek search to identify conserved folds. We empirically tested which window size would result in the highest number of significant hits using Foldseek. Trying a window size of 25, 50, 100, 150, and 200 amino acids, we found the most significant hits with a window size of 150. The sliding window step size was tested at 5, 10, 15, 20, and 25 amino acids. We used a step size of 20 amino acids for the final analysis as it resulted in the maximal number of protein species detected. This approach resulted in over 12,000 structural snippets of 150 residues in length, each overlapping by 130 residues with the preceding fragment and covering the whole folded human herpesvirus proteome. All structural fragments were compared pairwise with Foldseek, accounting for about 1.6 million tested combinations. We next assembled "structural contigs" of overlapping frames that passed the Foldseek threshold of an E value < 0.001. If a structural snippet matched another structural snippet and they were not overlapping in the same original protein, then the pair was deemed a "hit". If several structural snippets overlapped in sequential order, they were used to build larger domains. This analysis resulted in a list of structured areas, which we used to determine if a query protein has a different domain architecture than a structurally similar protein in this dataset. If a snippet of a query protein matched a different part of the query protein, we deemed it to have an "internal duplication". If a target snippet matched the query protein at multiple locations, we considered the query protein to have a "repetitive acquisition". If the query protein was structurally similar to a target protein but the query protein had at least one domain that was not found in the target protein, we deemed the protein to have a "domain addition". The associated script can be found in our HerpesFolds GitHub repository in the files "make_sliding_window_pdb.py", "Sliding_Window_Analysis.ipynb", "analyze_foldseek_sliding_window.py", and "analyze_sliding_window_domains.py" at https://github.com/QuantitativeVirology/Herpesfolds.

### Keyword analysis of Foldseek results

Most Foldseek searches against cellular databases resulted in multiple significant matches, each containing protein descriptions and matching protein identifiers. We filtered for the most enriched keywords to extract likely biological functions from these descriptions. The following terms, in addition to single characters, single and double digits, and punctuation, were filtered out as they are too generic:

crystal, a, after, allosteric, allosteric, alpha, an, and, angstrom, angstroms, antibody, apo, as, at, atomic, based, basic, beta, between, bound, by, by, c1, c-c, cell, chain, chi, class, class, coli, complex, complexed, complexes., compound, conserved, cryoem, cryo-em, crystal, c-terminal, de, dehydration, deletion, delta, design, different, discovery, domain, e., e.coli, edition, element, engineered, epsilon, escherichia, eta, fab, for, form, fragment, from, full, full-length, functional, gamma, herpes, herpesvirus, high, holoenzyme, ii, iii, implications, in, inhibitor, inhibitors, ion, iota, its, kappa, lambda, ligand, ligand-binding, loader, loop, low-ph, mode, molecule, monoclonal, motif, mu, mutant, natural, neutralizing, novel, novo, nu, of, omega, omicron, on, one, open, open, opposite, peptide, phi, pi, probable, protein, psi, region, resolution, reveal, rho, sigma, significance, site, specificity, structural, structure, structures, substrate, symmetry, tau, template-primer, the, the, therapeutic, theta, to, type, uncharacterized, unit, upsilon, using, variant, virus, with, xfel, xi, x-ray, zeta.

The used script can be found in our HerpesFolds GitHub repository at https://github.com/QuantitativeVirology/Herpesfolds in the files "Keyword_Frequency_SP_PDB.ipynb". This script was written with the assistance of a large language model.

### Reporting summary

Further information on research design is available in the Nature Portfolio Reporting Summary linked to this article.

## Data availability

The structure predictions and ChimeraX sessions generated in this study have been deposited in the Zenodo database under https://doi.org/10.5281/zenodo.13284140 at https://zenodo.org/records/13284140. Moreover, predictions and interactive clustering are available through a web interface at https://www.herpesfolds.org/herpesfolds and https://www.herpesfolds.org/herpesclusters.

## Code availability

The scripts are available at https://github.com/QuantitativeVirology/Herpesfolds, and a permanent reference is under https://doi.org/10.5281/zenodo.13991336 at https://zenodo.org/records/13991336.

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

## Acknowledgements

We would like to thank the Topf Lab at CSSB for help in the initial phases of this project, Milot Midarta and Martin Steinegger for help with Fold-seek and for providing links to Foldseek in HerpesFolds and Lisa Holm for advice in using DALI as a local installation. Finally, we thank Christian Löw for his help in evaluating if the BMRF2 protein cluster codes for functional transporters. This research was partly supported by the Maxwell computational resources operated at Deutsches Elektronen-Synchrotron DESY, Hamburg, Germany. Work in the Bosse lab was funded by the Deutsche Forschungsgemeinschaft (DFG, German Research Foundation) under Germany's Excellence Strategy EXC 2155—project no. 390874280, the DFG-funded RTG 2771 Humans and Microbes, project no. 453548970 (Bosse), the DFG-funded RTG 2887, project number 49735088, by the Wellcome Trust through a Colla-borative Award (209250/Z/17/Z) and the Leibniz ScienceCampus Inter-ACt, funded by the BWFGB Hamburg and the Leibniz Association (W75/2022) InterACt and "Hamburg- X Infektionsforschung". Moreover, the Bosse and Kaufer labs are both funded through the DFG Research Unit FOR5200 DEEP-DV (443644894) project BO 4158/5-1 and KA 3492/12-1 awarded to J.B.B. and B.B.K., respectively. B.K.K. is also funded by the ERC consolidator grant (ERC-CoG ENDo-HERPES, 101087480).

## Author contributions

T.K.S.: conceptualization, data curation, formal analysis, investigation, methodology, software, supervision, validation, visualization, writing—original draft, writing—review and editing. S.O.: data curation, software. S.S.: data curation, software. R.S.: data curation, software. B.K.: data curation, writing—original draft, writing—review and editing, funding acquisition, project administration. J.B.B.: conceptualization, data cura-tion, funding acquisition, methodology, project administration, super-vision, writing—original draft, writing—review and editing.

## Funding

## Competing interests

The authors declare no competing interests.

## Additional information

**Peer review information** *Nature Communications* thanks Arne Elofsson, Stephen Graham, and the other, anonymous, reviewer(s) for their con-tribution to the peer review of this work. A peer review file is available.

