## [Peer Review File · Nature Communications]

REVIEWER COMMENTS

Reviewer #1 (Remarks to the Author):

In this paper, the authors apply a structural clustering of proteins from different Herpes Viruses. As has been done in a number (of larger scale) studies recently, the author finds some clusters that are hard (or impossible) to detect using sequence alone. However, if this has any significant novelty, it is more doubtful.

Major:

The major problem with the paper is that any two different clustering methods will always create slightly different clustering depending on small differences in how clusters are generated.

First of all, there is no comparison with how the clustering on sequences (or Pfam domain) would have looked like. At the bare minimum, a comparison of all pairwise sequence similarities (using a HMM-HMM scoring method) should be performed. I would assume that most "distances" between pairs of proteins would be very strongly correlated while (as expected) a few pairs are only detected with one or the other of the methods. However, it should not always be assumed that structural similarity is "better" than sequence similarity - this depends from case to case. A better way to cluster the proteins would probably be based on Pfam.

One example where I would argue that sequence similarity is better is the small capsomere-interacting protein. Using a structural alignment method, they end up in different clusters, while (according to the authors) they are easily detected using sequence similarity. This is because these structures are mainly disordered, and the only ordered part is one helix - which (I assume) is too short to generate a significant structural similarity score. The author's claim that "Genetic drift may have led to structurally distinct features" is not correct; it is much more likely that this is caused by flexibility and not structurally distinct features.

Further, the claim that "the DNA polymerase catalytic subunit and DNA helicase/primase complex-associated protein exhibit similar folds" is surprising, as these proteins (according to the authors) "have some sequence similarity". This is clearly just one more example of homologs with different functions.

It is possible that the clustering should exclude short proteins - and disordered regions. Perhaps anything with less than ~100 well predicted residues should be excluded. Alternatively here the structural comparison should be combined with sequence similarity.

Minor:

The structural network clustering does not work on this page-

<https://www.bosse-lab.org/herpes-network/>

I get the following error: "Issue while parsing JSON: network. Please contact the page developer to get this issue resolved or check the browser console."

The way selection of models for evaluating quality is entirely ad hoc. The authors need to show that this is actually a useful method that uses some type of benchmark. Currently, only one example is shown - please provide data for all models.

Figure S2. It would be logical to use $\log(\text{E-value})$ for the plot and not the number

In summary:

The paper provides a clustering of herpes virus proteins using structural similarity. However, it is unclear if this provides any significant novel insights (not being an expert in herpes viruses, I leave this to such experts to judge). The major problem is that the authors assume that structural similarity is always relevant - but ignore the fact that (in particular for short/disorder proteins) structural variability can be large even for closely homologous proteins. Authors need to compare their clustering with existing methods to cluster AlphaFoldDB (look at papers by Steinegger, Orengo and others) with sequence-based and domain-based methods. I would assume that in general, all methods would give similar clustering and that most differences are based on exact choices of clustering methodology.

/Arne

Reviewer #1 (Remarks on code availability):

Sorry had no time to review the code

Reviewer #2 (Remarks to the Author):

The HerpesFold website is an incredibly useful resource that will be highly impactful (and highly cited) in future years. The authors describe how they generated this resource and then explore some of the features identified in the data, choosing to focus on domains conserved across herpesviruses and novel identification of potential homology with cellular proteins for a small subset of herpesvirus folds. The manuscript is clear and concise. Overall this study is very well executed and worthy of publication. However, I have some concerns about the structural equivalence studies that should be addressed, and a few suggestions where relatively modest additional effort would significantly improve the utility of the HerpesFold website for the community.

Major comments:

1. There is evidence that many more proteins are expressed by human herpesviruses than might be expected based purely on the 'canonical' map of ORFs (e.g. the HCMV non-canonical ORFs identified in <https://doi.org/10.1016/j.cell.2014.04.028>, the additional 6-frame translation open reading frames identified for HSV-1 in <https://doi.org/10.1016/j.celrep.2020.108235>, or the additional translation products and splice variants identified for HSV-1 in <https://doi.org/10.1038/s41467-020-15992-5>). Have the authors included these additional ORFs when making the HerpesFold database? What was the criteria used for inclusion/exclusion of these non-canonical ORFs? Could the authors comment on whether the ability to predict a high-quality structure for these additional ORFs could be used to assess whether these ORFs are likely to be functional in the context of infection?

2. The authors limited their predictions to three recycles. While this is the default for AlphaFold, Mirdita and colleagues (<https://doi.org/10.1038/s41592-022-01488-1>) showed that increasing the number of recycles improves prediction performance, especially for 'difficult' targets (see figures S5 and S6). The authors should re-run the predictions for proteins that have 'failed' the validation with additional recycles (up to 20), to see whether this can rescue a high-quality model.

3. As described, the choice of criteria for structural equivalence is not compelling. The plot of Foldseek E value vs Dali Z-score in Figure S2 does not convince me that they have established robust and reliable metrics for equivalence. The data may be easier to interpret if the horizontal axis of (A) was on a logarithmic scale, but the data in the zoomed-in panel (B) are only very poorly correlated, if at all. The choice of '0.001' as a Foldseek E value for equivalence thus seems arbitrary. In line 155-156 the authors state that "the results from both algorithm [DALI and Foldseek] were similar", but given the data presented I can't see how they can justify this conclusion.

FoldSeek has two different modes (3Di/AA and TM-align) – the authors do not explicitly specify which was used (presumably 3Di/AA, given the reporting of E values) nor do they justify this choice. This is relevant as the TM-align option of FoldSeek is reported to outperform the 3Di/AA mode in terms of sensitivity (<https://doi.org/10.1038/s41587-023-01773-0>). Secondly, if the authors are to use the 3Di/AA mode of Foldseek, why do they use the E value rather than probability, which has been calibrated to false positive probability on the SCOPe benchmark and thus could be used to impose a 'meaningful' false discovery rate? Did the authors perform any experiments to determine the false positive/negative rate for their definition of structural equivalence?

Given that a large proportion of the biological insight presented in this manuscript derives from the identification of equivalent domains/proteins across homologues, robust justification of the criteria for defining equivalence is crucial. There are examples in the Herpesfold database where established equivalencies are not reproduced using the analysis pipeline presented (for example, HSV-1 UL51 and VZV_ORF7 do not cluster with HCMV_UL71 [rows 48 and 86 of supplemental file 2 sheet 2], despite these being presented as the same structural group [row 125 of same excel file]). Such inconsistencies should be flagged more clearly in the supplementary material and on the herpesfold website.

4. The authors have not considered signal sequences or trans-membrane regions of single-span transmembrane proteins when performing predictions. These regions would be trivial to remove automatically via analysis of the sequences using DeepTMHMM (<https://www.biorxiv.org/content/10.1101/2022.04.08.487609v1>) and re-calculation of the models for the relevant extracellular and intracellular domains. Example python code to do this for single-span transmembrane proteins is attached to this review. The authors present their database as tool for accelerating experiment design by wet-lab scientists, who might not have the structural familiarity to understand that some regions of the structural model would not be present in the mature protein and/or that domains would be separated by a membrane. I would strongly suggest that they update their database to include predictions of extracellular and intracellular domains for secreted and single-pass transmembrane proteins.

5. The potential identification of HSV US2 as a dihydrofolate reductase is interesting, but it is hard to evaluate using the data presented. In particular, Figure S4 is very unclear when it comes to assessing the conservation of the conserved catalytic residues. The authors should include additional zoomed views of the relevant residues, with atoms coloured by element, plus a structure-based sequence alignment (e.g. Chimera Match>Align) where catalytic residues are highlighted. Additionally, the authors state that the ChimeraX scene file for the illustration presented in Figure S4 could be found on the herpesfold website but I could not find it. This should be clarified.

Minor comments:

Line 54-55: Structure-based phylogeny has been used in virology for many years, see for example the pioneering work on Dennis Bamfield and David Stuart in this area (reviewed in <https://doi.org/10.1146/annurev-biochem-060910-095130>; see also <https://doi.org/10.1093/ve/veaa003>). It's more accurate to say that AlphaFold has vastly expanded the pool of sequences that can be interrogated by structure-based phylogeny.

Line 265: RMSD values must always be accompanied by the number of atoms (presumably C α atoms) for which they are calculated. Please insert number of equivalent residues for each point in the paper where RMSDs are quoted.

Figure 3B: It would be predicted that conserved DNA binding proteins would have conserved basic patches on their surface. Please provide surface electrostatics of the LANA-like homologues as a supplemental figure, in support of the hypothesis that these proteins have conserved DNA-binding functions across herpesviruses. If there is not conservation of surface electrostatic features, this should be commented on.

Reviewer #3 (Remarks to the Author):

State-of-the-art machine learning tools are, for the first time, permitting accurate protein structure prediction from primary sequence alone. These tools are ushering a new era of rapid and scalable computational investigation of protein function and evolution. However, thus far these approaches have

yet to be systematically applied to viral proteins. In this study, Soh et al., use ColabFold and AlphaFold (two leading, and related, approaches to protein structure prediction) to predict structures for the entire proteome of all nine human herpesviruses. These structures were then analysed in detail, largely using the powerful Foldseek tool, to understand protein function and relatedness. This analysis included some innovative approaches to examine domain duplication events and how these may propagate throughout a viral genome. This work has three key strengths: i) it provides a template of how state-of-the-art protein structure prediction can be applied to virology ii) the predicted structures are available via a website, alongside various quality metrics, this is a resource for the herpesvirus research community, iii) the analysis, in and of itself, give biological insight on herpesviruses (e.g. examples of cellular protein exaptation) which, again, will be of interest to the research community. I am, therefore, in favour of the publication of this work. I would, however, recommend some changes that I think would improve the work.

Major suggested edits.

- Throughout the manuscript the figures need some redesign to improve clarity and intuitiveness. I provide a few examples, but would suggest reassessing all of the figures. Figure 1A – why not use protein structures to illustrate workflow (not scribbled lines), figure 1 F, G, H, these images are too small to make use of, are they needed? Figure 2 – virtually no labels and those that are there are too small to read. Can you not relate the highlighted clusters (B C D) to the overall cluster map (A). In some figures (e.g. 5) the overlaid structures are too ‘busy’ to extract any information from or, as in Fig 4 B, the relevant color-coded regions are almost impossible to see. Essentially, the current data presentation presents a barrier to readers fully appreciating the value of the work.

- The authors have performed quite a lot of analysis on disordered regions in the viral genomes using pLDDT scores as a proxy measure of disorder. However, I feel like this element of the work was under-cooked. Could the authors not expand on this, providing examples of highly disordered proteins and those that contain disordered regions (e.g. N or Cterm tails), example images and quantification would be good. Also, how does the frequency of protein disorder in herpesviruses compare to other organisms?

Minor suggested edits.

- I believe the ‘e-value’ in Foldseek is short for ‘expect-value’ not ‘extreme value’. Also I think the e-value is a sequence alignment metric (in this case for structurally aligned sequence) and is influenced by database size. Whilst it is a good metric for evaluating Foldseek outputs others (e.g. Bit score and LDDT) are more representative of structural similarity and, therefore, may better relate to the Z-score from DALI. The authors may want to consider some edits and/or caveats around this?

- Whilst I have not treated the www.bosse-lab.org/herpesfolds/ website as fully part of this peer review (after all it will likely naturally change over time), I may argue that it doesn’t quite “serve as a reference that can translate the complicated herpesvirus nomenclature for the expert and non-expert alike”. I am non-expert in herpes virology and it is still a little tricky to navigate – cartoon representations of the viral particle and/or genome might be a useful orientation?

- Line 197 “than” should this be “that”?

- Figure 1D and E, axis labels or numbers are inconsistent. Label states % pLDDT whereas values are all in fractions of 1 (where I would expect values between 0-100%).

- Figure 4 – might it be valuable to use a linear representation of a herpes virus genome to illustrate the

various domain duplication events occurring, this may be intuitive and complement the current figure.
- Figure 4C – shades of red are almost indistinguishable.

Reviewer #3 (Remarks on code availability):

The code is well documented with appropriate readme files and I am confident that it would be possible to reproduce the work.

However, I don't believe the HerpesFold website has a 'download all structures' function, which may be useful. Furthermore, access to the files is dependent on the website remaining active. I would therefore recommend depositing all structures in a repository like Zenodo. This would allow readers to download all structures and increase data security.

Point by point response:

We sincerely thank all reviewers for their critical reading of our manuscript and their insightful suggestions. Below, is our point-by-point response to the reviewers' comments, highlighted in blue.

We have reformatted the manuscript to fit the Nature Communications formatting instructions. The content changes have been highlighted in yellow.

Reviewer #1 (Remarks to the Author):

In this paper, the authors apply a structural clustering of proteins from different Herpes Viruses. As has been done in a number (of larger scale) studies recently, the author finds some clusters that are hard (or impossible) to detect using sequence alone. However, if this has any significant novelty, it is more doubtful.

We thank reviewer 1 for the critical evaluation of our manuscript and important suggestions. We provide a detailed response below.

Major:

The major problem with the paper is that any two different clustering methods will always create slightly different clustering depending on small differences in how clusters are generated.

First of all, there is no comparison with how the clustering on sequences (or Pfam domain) would have looked like. At the bare minimum, a comparison of all pairwise sequence similarities (using a HMM-HMM scoring method) should be performed. I would assume that most "distances" between pairs of proteins would be very strongly correlated while (as expected) a few pairs are only detected with one or the other of the methods. However, it should not always be assumed that structural similarity is "better" than sequence similarity - this depends from case to case. A better way to cluster the proteins would probably be based on Pfam.

We have repeated the clustering using MMseqs2 and hhblits and have included the results in Supplementary Data 2 Sheet 2 and 3, respectively. The clusters we identified using these algorithms are consistent with current literature. For example, the serine/threonine protein kinase cluster was identified, which includes the UL13 ortholog from each species and the US3 ortholog in each alphaherpesvirus [DOI: 10.1128/JVI.01369-10]. Another example is the portal proteins, which are highly conserved and found in every species [Human Herpesviruses: Biology, Therapy, and Immunoprophylaxis]. Many more verify that we applied the software correctly. Importantly, the novel connections that we identified through structural alignment are missing, e.g. HCMV UL112-113 with HSV-1 UL4 and KSHV ORF2 with HSV-1 US2 (See line 168, 329, and 395). These results support the notion that, at least in the case of herpesvirus subfamilies which have diverged hundreds of millions of years ago, structure-based clustering is advantageous.

One example where I would argue that sequence similarity is better is the small capsomere-interacting protein. Using a structural alignment method, they end up in different clusters, while (according to the authors) they are easily detected using sequence similarity. This is because these structures are mainly disordered, and the only ordered part is one helix - which (I assume) is too short to generate a significant structural similarity score. The author's claim that "Genetic drift may have led to structurally distinct

features" is not correct; it is much more likely that this is caused by flexibility and not structurally distinct features.

The above-outlined sequence-based searches provided the same clustering for alpha- and beta-herpesviruses as our structure-based search. These results are consistent with our conclusion that the subfamilies have diverged. We would also like to point out that the predicted structures are different in the number and length of alpha-helices. If the Foldseek-based clustering approach was random due to the disordered regions, it is highly unlikely it would have separated these proteins into exactly these subfamilies.

Further, the claim that "the DNA polymerase catalytic subunit and DNA helicase/primase complex-associated protein exhibit similar folds" is surprising, as these proteins (according to the authors) "have some sequence similarity". This is clearly just one more example of homologs with different functions.

By HHblits sequence clustering, the DNA polymerase catalytic subunit forms a single cluster and the DNA helicase/primase complex-associated protein form a distinct cluster. In contrast, Kazlauskas and Venclovas [DOI: 10.1093/bioinformatics/btu204] have shown through "Systematic iterative sequence searches using Jackhmmer" that the HSV-1 DNA helicase/primase (UL8) C-terminus has sequence similarity to the polymerase (UL30) C-terminus. Since structure is dependent on sequence and they have partial sequence similarity, it is not surprising that there is partial structural similarity. Since all of the viruses have both genes, we agree with the reviewer. It suggests that they are paralogs that have diverged in function. The text has been adjusted for clarity, line 171.

It is possible that the clustering should exclude short proteins - and disordered regions. Perhaps anything with less than ~100 well predicted residues should be excluded. Alternatively here the structural comparison should be combined with sequence similarity.

We respectfully disagree. In our opinion, proteins should only be excluded if their predictions are of low confidence. Some viral proteins are less than 100 residues and still well-predicted. When choosing our approach, we wanted to avoid an arbitrary cut-off. Foldseek uses a local structural alignment algorithm that should avoid the problems of flexible disordered regions. Accordingly, the small capsomere-interacting protein is clustered correctly (see above).

Minor:

The structural network clustering does not work on this page-<https://www.bosse-lab.org/herpes-network/>. I get the following error: "Issue while parsing JSON: network. Please contact the page developer to get this issue resolved or check the browser console."

We appreciate the heads-up. From outside our campus network, the Drugst.One plugin took a while to load. We have replaced it with a Sigma.js interface that runs much smoother (<https://www.bosse-lab.org/herpesclusters/>)

The way selection of models for evaluating quality is entirely ad hoc. The authors need to show that this is actually a useful method that uses some type of benchmark. Currently, only one example is shown - please provide data for all models.

We respectfully disagree. The evaluation of the model quality was applied uniformly, not *ad hoc*. All models had to pass all 3 criteria to be deemed good quality, line 86. The pTM criteria was determined through ROC analysis, and we chose to include a “consistency” criteria between the models to quantify how complete the optimization of each model was by AlphaFold. The “consistency” scores fit a Gaussian distribution with an R^2 of 94% and 96%, which is indicative of a normal distribution and outliers. Since AlphaFold will always generate a structure, quantitative thresholds are needed. We have set three based on statistical analyses of first-principles approaches. While it is true that we have not benchmarked these values to demonstrate that they improve the model accuracy, doing so would require solving the structure of multiple proteins that failed the quality scores, which is outside the scope of this work. The pTM threshold is arguably benchmarked since the viral proteins are known to fold stably and a scrambled sequence is generally accepted to not fold. The ROC area under the curve is 91.3%, and >90% is “considered outstanding” [DOI: 10.1097/JTO.0b013e3181ec173d]. The distribution of the quality scores for all models are also shown in Fig S1. All models are now available for download at <https://zenodo.org/records/13284140>.

Figure S2. It would be logical to use $\log(\text{E-value})$ for the plot and not the number

Due to the feedback from the other reviewers, we have chosen to remove Figure S2.

In summary:

The paper provides a clustering of herpes virus proteins using structural similarity. However, it is unclear if this provides any significant novel insights (not being an expert in herpes viruses, I leave this to such experts to judge). The major problem is that the authors assume that structural similarity is always relevant - but ignore the fact that (in particular for short/disorder proteins) structural variability can be large even for closely homologous proteins. Authors need to compare their clustering with existing methods to cluster AlphaFoldDB (look at papers by Steinegger, Orengo and others) with sequence-based and domain-based methods. I would assume that in general, all methods would give similar clustering and that most differences are based on exact choices of clustering methodology.

We thank the reviewer for the suggestion and also tested the Foldseek easy-cluster algorithm, which adds a 90% (recommended) fraction of aligned (covered) residues requirement. This setting resulted in much smaller clusters (only a few contained one protein from each species) and did not identify the paralog families or the novel hits we found. Since our goal is to identify distant connections between proteins as well as similar protein folds due to convergent evolution, this setting is not useful in our case. Moreover, many relations are missed using sequence-based approaches such as the HSV-1 UL4 and HCMV UL112-113 cluster. Here, literature [DOI: 10.1128/jvi.74.1.523-528.2000, DOI: 10.1016/j.celrep.2022.110469] clearly suggests that they have similar subcellular localization and behavior, and we point out in this manuscript that the common localization might be due to a common function. See line 182.

/Arne

Reviewer #1 (Remarks on code availability):

Sorry had no time to review the code

Reviewer #2 (Remarks to the Author):

The HerpesFold website is an incredibly useful resource that will be highly impactful (and highly cited) in future years. The authors describe how they generated this resource and then explore some of the features identified in the data, choosing to focus on domains conserved across herpesviruses and novel identification of potential homology with cellular proteins for a small subset of herpesvirus folds. The manuscript is clear and concise. Overall, this study is very well executed and worthy of publication. However, I have some concerns about the structural equivalence studies that should be addressed, and a few suggestions where relatively modest additional effort would significantly improve the utility of the HerpesFold website for the community.

We thank reviewer 2 for the positive evaluation and important suggestions. We provide a detailed response below.

Major comments:

1. There is evidence that many more proteins are expressed by human herpesviruses than might be expected based purely on the 'canonical' map of ORFs (e.g. the HCMV non-canonical ORFs identified in <https://doi.org/10.1016/j.cell.2014.04.028>, the additional 6-frame translation open reading frames identified for HSV-1 in <https://doi.org/10.1016/j.celrep.2020.108235>, or the additional translation products and splice variants identified for HSV-1 in <https://doi.org/10.1038/s41467-020-15992-5>). Have the authors included these additional ORFs when making the HerpesFold database? What was the criteria used for inclusion/exclusion of these non-canonical ORFs? Could the authors comment on whether the ability to predict a high-quality structure for these additional ORFs could be used to assess whether these ORFs are likely to be functional in the context of infection?

To set an objective criteria of what to include, we used the respective UniProt proteomes or reference genomes (for HHV-6A, -6B, and -7 since the UniProt is not complete) for the viral species and indicated strain. We agree that further work on "non-canonical" ORFs would be highly interesting. Unfortunately, experimental data on "non-canonical" ORF expression is not available for all human herpesviruses and we did not want to include purely hypothetical ORFs generated by *in-silico* 6-frame translation. That said, we are very interested in such a study even though it is out of the scope of this manuscript. We make a brief reference to this point in line 382.

2. The authors limited their predictions to three recycles. While this is the default for AlphaFold, Mirdita and colleagues (<https://doi.org/10.1038/s41592-022-01488-1>) showed that increasing the number of recycles improves prediction performance, especially for 'difficult' targets (see figures S5 and S6). The authors should re-run the predictions for proteins that have 'failed' the validation with additional recycles (up to 20), to see whether this can rescue a high-quality model.

We agree and thank reviewer 2 for this suggestion. We have rerun the ColabFold 3x recycles models that failed the quality scores again with ColabFold 20x recycles. If both ColabFold 20x recycles and the AlphaFold model passed the quality score, the model with the higher pTM was used for the structural alignment. See line 91.

3. As described, the choice of criteria for structural equivalence is not compelling. The plot of Foldseek E value vs Dali Z-score in Figure S2 does not convince me that they have established robust and reliable

metrics for equivalence. The data may be easier to interpret if the horizontal axis of (A) was on a logarithmic scale, but the data in the zoomed-in panel (B) are only very poorly correlated, if at all. The choice of '0.001' as a Foldseek E value for equivalence thus seems arbitrary. In line 155-156 the authors state that "the results from both algorithm [DALI and Foldseek] were similar", but given the data presented I can't see how they can justify this conclusion.

We chose 0.001 as a Foldseek E-value threshold because it has been used before in similar studies [DOI: 10.1128/mbio.00408-23]. We also used a DALI Z-score of 8 because the DALI authors recommend it and has been used as a standard [DOI: 10.1007/978-1-0716-0270-6_3]. We agree that comparing both thresholds is not straightforward, and we do not want to establish new comparative thresholds. Given the feedback from the reviewers, we have decided to remove this figure and discussion.

FoldSeek has two different modes (3Di/AA and TM-align) – the authors do not explicitly specify which was used (presumably 3Di/AA, given the reporting of E values) nor do they justify this choice. This is relevant as the TM-align option of FoldSeek is reported to outperform the 3Di/AA mode in terms of sensitivity (<https://doi.org/10.1038/s41587-023-01773-0>). Secondly, if the authors are to use the 3Di/AA mode of Foldseek, why do they use the E value rather than probability, which has been calibrated to false positive probability on the SCOPe benchmark and thus could be used to impose a 'meaningful' false discovery rate? Did the authors perform any experiments to determine the false positive/negative rate for their definition of structural equivalence?

We tried TAlign in addition to the 3Di+AA Gotoh-Smith-Waterman algorithm for structural alignment and used the probability as well as the E-value to define significant connections. The 3Di+AA Gotoh-Smith-Waterman algorithm with an E-value or probability threshold generated multiple clusters that matched known protein families. In contrast, using TAlign with either value resulted in few clusters that contain 100s of proteins. Increasing the threshold resulted in fewer proteins being included in the cluster rather than breaking the cluster into smaller sub-clusters. The 3Di+AA Gotoh-Smith-Waterman algorithm with a probability threshold resulted in a similar result to the E-value. Since the E-value has been used for a similar study [DOI: 10.1128/mbio.00408-23], we decided to continue with the E-value. See Supplementary Data 2, Sheet 7 and 8. See line 442.

Given that a large proportion of the biological insight presented in this manuscript derives from the identification of equivalent domains/proteins across homologues, robust justification of the criteria for defining equivalence is crucial. There are examples in the Herpesfold database where established equivalencies are not reproduced using the analysis pipeline presented (for example, HSV-1 UL51 and VZV_ORF7 do not cluster with HCMV_UL71 [rows 48 and 86 of supplemental file 2 sheet 2], despite these being presented as the same structural group [row 125 of same excel file]). Such inconsistencies should be flagged more clearly in the supplementary material and on the herpesfold website.

We agree and have annotated the table correspondingly. There were three examples where established groups were missing an expected protein because the respective structure prediction failed a quality score. These are now also pointed out in the text and on the website, see 353.

4. The authors have not considered signal sequences or trans-membrane regions of single-span transmembrane proteins when performing predictions. These regions would be trivial to remove automatically via analysis of the sequences using DeepTMHMM

(<https://www.biorxiv.org/content/10.1101/2022.04.08.487609v1>) and re-calculation of the models for the relevant extracellular and intracellular domains. Example python code to do this for single-span transmembrane proteins is attached to this review. The authors present their database as tool for accelerating experiment design by wet-lab scientists, who might not have the structural familiarity to understand that some regions of the structural model would not be present in the mature protein and/or that domains would be separated by a membrane. I would strongly suggest that they update their database to include predictions of extracellular and intracellular domains for secreted and single-pass transmembrane proteins.

We agree that the presence of the signal peptide and transmembrane domain(s) could be confusing. For some proteins, the signal peptide was predicted to be the majority of the protein (such as KSHV K12). Since these would be unusable for structural alignment after removing the signal peptide, we continued to use the full-length protein for structural alignment but included a second prediction on the individual web pages where the signal peptide sequence was removed before the prediction. Proteins with a predicted transmembrane domain but not a signal peptide have the transmembrane domains listed on the webpage. We now also explicitly state the transmembrane domain as predicted by DeepTMHMM. An example can be found at: https://www.bosse-lab.org/HCMV_RL11/.

5. The potential identification of HSV US2 as a dihydrofolate reductase is interesting, but it is hard to evaluate using the data presented. In particular, Figure S4 is very unclear when it comes to assessing the conservation of the conserved catalytic residues. The authors should include additional zoomed views of the relevant residues, with atoms coloured by element, plus a structure-based sequence alignment (e.g. Chimera Match>Align) where catalytic residues are highlighted. Additionally, the authors state that the ChimeraX scene file for the illustration presented in Figure S4 could be found on the herpesfold website but I could not find it. This should be clarified.

We added zoomed in views to make our point clearer. We also added a sequence based and structure-based sequence alignments to Supplementary Fig. 5 (the new designation for Figure S4). Moreover we modified the text at line 294. The corresponding ChimeraX files can be found on our Zenodo page <https://zenodo.org/records/13284140>.

Minor comments:

Line 54-55: Structure-based phylogeny has been used in virology for many years, see for example the pioneering work on Dennis Bamfield and David Stuart in this area (reviewed in <https://doi.org/10.1146/annurev-biochem-060910-095130>; see also <https://doi.org/10.1093/ve/veaa003>). It's more accurate to say that AlphaFold has vastly expanded the pool of sequences that can be interrogated by structure-based phylogeny.

We thank the reviewer for this suggestion and apologize for the oversight. The sentence has been reworded and the references are now included at line 50.

Line 265: RMSD values must always be accompanied by the number of atoms (presumably C α atoms) for which they are calculated. Please insert number of equivalent residues for each point in the paper where RMSDs are quoted.

We apologize for the oversight. In Figure 4D, the RMSD used 91 alpha carbons, in Figure 5H, the RMSD used 205 alpha carbons, and in Supplementary Fig. 3 the RMSD used 310 alpha carbons. This has been added to the manuscript at line 118, 285, and the figure legends.

Figure 3B: It would be predicted that conserved DNA binding proteins would have conserved basic patches on their surface. Please provide surface electrostatics of the LANA-like homologues as a supplemental figure, in support of the hypothesis that these proteins have conserved DNA-binding functions across herpesviruses. If there is not conservation of surface electrostatic features, this should be commented on.

We agree and used AlphaFold 3 to predict the LANA-like proteins with DNA. They are included as Supplementary Fig. 4 and a discussion is included at line 190.

Reviewer #3 (Remarks to the Author):

State-of-the-art machine learning tools are, for the first time, permitting accurate protein structure prediction from primary sequence alone. These tools are ushering a new era of rapid and scalable computational investigation of protein function and evolution. However, thus far these approaches have yet to be systematically applied to viral proteins. In this study, Soh et al., use ColabFold and AlphaFold (two leading, and related, approaches to protein structure prediction) to predict structures for the entire proteome of all nine human herpesviruses. These structures were then analysed in detail, largely using the powerful Foldseek tool, to understand protein function and relatedness. This analysis included some innovative approaches to examine domain duplication events and how these may propagate throughout a viral genome. This work has three key strengths: i) it provides a template of how state-of-the-art protein structure prediction can be applied to virology ii) the predicted structures are available via a website, alongside various quality metrics, this is a resource for the herpesvirus research community, iii) the analysis, in and of itself, give biological insight on herpesviruses (e.g. examples of cellular protein exaptation) which, again, will be of interest to the research community. I am, therefore, in favour of the publication of this work. I would, however, recommend some changes that I think would improve the work.

We thank reviewer 3 for the positive evaluation and suggestions. We provide a detailed response below.

Major suggested edits.

- Throughout the manuscript the figures need some redesign to improve clarity and intuitiveness. I provide a few examples, but would suggest reassessing all of the figures. Figure 1A – why not use protein structures to illustrate workflow (not scribbled lines), figure 1 F, G, H, these images are too small to make use of, are they needed? Figure 2 – virtually no labels and those that are there are too small to read. Can you not relate the highlighted clusters (B C D) to the overall cluster map (A). In some figures (e.g. 5) the overlaid structures are too ‘busy’ to extract any information from or, as in Fig 4 B, the relevant color-coded regions are almost impossible to see. Essentially, the current data presentation presents a barrier to readers fully appreciating the value of the work.

We appreciate these suggestions and have tried to integrate them. We have used a protein structure instead of a cartoon representation of a folded protein in Fig. 1. Concerning the website screenshots we are trying to entice the reader to use the website. We believe it makes the structures more accessible than the download all button that requires searching through 100’s of GB of data. While Figure 2 is ineligible in a printed version, the vector graphic is readable. On a computer, one can zoom in and look for their protein of interest if they do not wish to use the website. We wanted to use Figure 1A as an overview of all of the data, which is easier to absorb than the chart in Supplementary Data 2 Sheet 5. With 9 species, we tried to color code the manuscript to make the figures consistent. Figure 4C now has more distinct shades of red to keep with the red color that was given to HCMV.

- The authors have performed quite a lot of analysis on disordered regions in the viral genomes using pLDDT scores as a proxy measure of disorder. However, I feel like this element of the work was under-cooked. Could the authors not expand on this, providing examples of highly disordered proteins and those that contain disordered regions (e.g. N or Cterm tails), example images and quantification would be good. Also, how does the frequency of protein disorder in herpesviruses compare to other organisms?

To expand on the analysis of the disordered regions, Supplementary Fig. 2 has been added. It illustrates the percentage of the proteome, by virus or the consolidate proteome, that has a structured or disordered

domain. In addition, it shows an example of a C-terminal and N-terminal disordered protein. The distribution of proteins based on their percentage of structured or unstructured domains can be found in Figure 1 D and E. We have also included a comparison to the MobiDB and the literature as described in line 103.

Minor suggested edits.

- I believe the 'e-value' in Foldseek is short for 'expect-value' not 'extreme value'. Also I think the e-value is a sequence alignment metric (in this case for structurally aligned sequence) and is influenced by database size. Whilst it is a good metric for evaluating Foldseek outputs others (e.g. Bit score and LDDT) are more representative of structural similarity and, therefore, may better relate to the Z-score from DALI. The authors may want to consider some edits and/or caveats around this?

<https://doi.org/10.1038/s41587-023-01773-0> states that “*E* values are calculated using an extreme-value score distribution...”, suggesting that the authors wish to define it as “extreme-value”. Still, we have changed it to match the NIH/BLAST definition of E-value as “Expect value” in line 444.

As described above, we chose 0.001 as a Foldseek E-value threshold because it was used recently in a similar study [DOI: 10.1128/mbio.00408-23]. A DALI Z-score of 8 as threshold was chosen as it is recommended by the authors [DOI: 10.1007/978-1-0716-0270-6_3]. Given the feedback from the reviewers concerning the comparison between the Z-score and E-value, we have removed this figure.

- Whilst I have not treated the www.bosse-lab.org/herpesfolds/ website as fully part of this peer review (after all it will likely naturally change over time), I may argue that it doesn't quite “serve as a reference that can translate the complicated herpesvirus nomenclature for the expert and non-expert alike”. I am non-expert in herpes virology and it is still a little tricky to navigate – cartoon representations of the viral particle and/or genome might be a useful orientation?

We designed the website in a way that it serves as a “cheat sheet” for herpes-virologists, where one can search for one viral protein and find the related proteins, their names, as well as their structures. Since the nomenclature of human herpesviruses is highly convoluted, this is a very helpful feature that we use daily in our own work and the community has given supportive feedback on. We agree that a graphical display would be great to illustrate where proteins are located in the virion or in the infected cell, especially for non-herpes researchers. Unfortunately, this information is not available for a large fraction of the included proteins. In addition, genome structure is not conserved, with some species having 2 distinct unique regions, which are flanked by repeat regions and can flip in orientation relative to each other, while others are a single linear genome. Displaying a common genome orientation is therefore also not possible. While conceptually challenging, we are open to suggestions concerning the user interface and user experience and will continue to work on the website to make it more accessible for all scientists.

- Line 197 “than” should this be “that”?

We rephrased this sentence for clarity. See line 482.

- Figure 1D and E, axis labels or numbers are inconsistent. Label states % pLDDT whereas values are all in fractions of 1 (where I would expect values between 0-100%).

The axes were updated.

- Figure 4 – might it be valuable to use a linear representation of a herpes virus genome to illustrate the various domain duplication events occurring, this may be intuitive and complement the current figure.

Unfortunately, the herpes genomes are organized very differently in different species. A point we make is that the location in the genome is different between species, and this suggests multiple integration events instead of a common ancestor.

- Figure 4C – shades of red are almost indistinguishable.

We apologize and changed the colors.

Reviewer #3 (Remarks on code availability):

The code is well documented with appropriate readme files and I am confident that it would be possible to reproduce the work.

However, I don't believe the HerpesFold website has a 'download all structures' function, which may be useful. Furthermore, access to the files is dependent on the website remaining active. I would therefore recommend depositing all structures in a repository like Zenodo. This would allow readers to download all structures and increase data security.

We thank reviewer 3 for this very positive assessment of our website. We uploaded the data to Zenodo: <https://zenodo.org/records/13284140>

REVIEWER COMMENTS

Reviewer #1 (Remarks to the Author):

I do thank the authors to have improved the manuscript and making the website much faster.

However, I do believe that there are a couple of outstanding issues that remain,

Clustering is dependent on cutoffs, i.e., to just say that a cluster is found by one method and not another is not sufficient. This needs to be further analyzed; at the bare minimum, for each pair reported to be found by one but not the other method, the pairwise scores should be reported.

Secondly, as the authors respond to my suggestion about excluding unreliable regions for clustering, the answer "We respectfully disagree. In our opinion, proteins should only be excluded if their predictions are of low confidence". However, from what I can see no exclusion of low-confidence regions ($pLDDT < 0.7$ or 0.5) is done. In case this is correct, I suggest that the author redo the clustering and exclude low-confidence regions as they themselves suggest.

Reviewer #2 (Remarks to the Author):

The authors have addressed most of my comments. Some minor issues that would need clarification before publication are as follows:

1. In response to my "major comment 1", the authors state that they included a reference to non-canonical ORFs in line 382 but I can't see any such text. Can they please confirm the statement that has been included.
2. I don't see code in the supplied github repository for the TMHMM analysis as included in Supplemental data file 1 sheet 6 (and on the website to generate signal peptide free structures). The authors should clarify the availability of this code.
3. On line 293, the authors have included the PDB reference to entry 3EIG but have not included a citation to the primary literature reference (Volpato, J.P., Yachnin, B.J., Blanchet, J., Guerrero, V., Poulin, L., Fossati, E., Berghuis, A.M., Pelletier, J.N. (2009) *J Biol Chem* 284: 20079-20089). This is exceedingly disrespectful to our structural colleagues - they are to thank for all the experimental data that underpins the success of AlphaFold and related methodologies! The relevant primary literature reference should be included here and any other time a PDB entry is referenced.

Reviewer #2 (Remarks on code availability):

I would have preferred if the authors would have also included the scripts used to perform the structural predictions, in addition to the analysis scripts. I can't see that readers could repeat their analysis on

another family of viruses without writing such scripts/code themselves, which seems like a missed opportunity. The code supplied for post-prediction structure analysis seems appropriately documented from a brief review.

Reviewer #3 (Remarks to the Author):

I thank the authors for considering my comments and amending their manuscript. Broadly, I remain enthusiastic about the work and its value to the community.

I am, however, a little disappointed that the authors did not do more to increase the visual accessibility of their work, I expect there were creative solutions to some of these problems. For example, scrutiny of the webpage screen shots in Figure 1 require zooming in, at which point they're revealed to be quite low resolution (not a good advert for the web resource). In Figure 2A, the clusters have no discernible labels, consequently, they are simply collections of colorful dots. In fact, each dot in this plot has a label, but these are invisible at standard view (i.e. it is unlikely the reader will know that they are there) and they only become legible at maximum zoom. This is before we consider the color-blind compatibility.

Ultimately, this comes down to how the authors want to communicate their work and not the quality of the scientific content of the manuscript. Therefore, if the authors are satisfied with this presentation I do not see this as a barrier to publication. However, if the work is accepted, I would recommend these issues are returned to at the copyediting stage.

Reviewer #3 (Remarks on code availability):

I remain satisfied with the code and data availability, particularly as the authors now provide all data in a Zenodo repository.

Point by point response:

We again thank all reviewers for their continued critical input. Below, is our point-by-point response to the reviewers' comments, highlighted in blue.

Reviewer #1 (Remarks to the Author):

I do thank the authors to have improved the manuscript and making the website much faster.

However, I do believe that there are a couple of outstanding issues that remain,

Clustering is dependent on cutoffs, i.e., to just say that a cluster is found by one method and not another is not sufficient. This needs to be further analyzed; at the bare minimum, for each pair reported to be found by one but not the other method, the pairwise scores should be reported.

We have added sheet 10 to Supplementary Data 2 to include the suggested data. This sheet lists all protein pairs exported by HHblits or Foldseek and shows the significant scores side by side. Furthermore, the significant protein pairs are listed along with the pairwise scores, and they are explicitly separated by which pairs were found by only HHblits, only Foldseek, or both. See lines 217-228. The HHblits cutoff is the recommended E-value \$< 0.001\$ (<https://toolkit.tuebingen.mpg.de/tools/hhblits>).

We amended the manuscript with a discussion of the differences found. In brief, we found that only structural clustering uncovered important novel biological findings such as the UL112-113 cluster as well as the LANA cluster. In both cases sequence conservation was too low to be detected otherwise. However, sequence-based clustering was helpful in cases where the structural predictions failed. Therefore, we have now added an option on our website at <https://www.herpesfolds.org/herpesclusters> where one can compare and merge the sequence- and structure-based clustering in a straightforward GUI. We thank the reviewer for hinting at this point.

Secondly, as the authors respond to my suggestion about excluding unreliable regions for clustering, the answer "We respectfully disagree. In our opinion, proteins should only be excluded if their predictions are of low confidence". However, from what I can see no exclusion of low-confidence regions ($pLDDT < 0.7$ or 0.5) is done. In case this is correct, I suggest that the author redo the clustering and exclude low-confidence regions as they themselves suggest.

We have now also included a separate clustering of protein fragments where the low-confidence region(s) were removed prior to running them in Foldseek. To do so, we isolated regions with a \$pLDDT > 0.7\$ for each viral protein. If adjacent regions were connected by a disordered region, i.e. \$pLDDT < 0.7\$, that was shorter than 100 residues, the adjacent regions along with the disordered region were counted as a single domain, a cutoff suggested in the previous review round. Subsequently, Foldseek clustering was repeated and compared to the clustering of full-length proteins (Supplementary Data 2 sheet 9). While some new connections were identified, there were many connections between proteins with little tertiary structure. Moreover, the identified connections often appeared to make little sense from a virological point of view.

For this reason, we decided to focus on the clustering using full-length predictions but include the suggested analysis in the supplementary data. Details on the identified proteins can be found at lines 207-216.

Reviewer #2 (Remarks to the Author):

The authors have addressed most of my comments. Some minor issues that would need clarification before publication are as follows:

1. In response to my "major comment 1", the authors state that they included a reference to non-canonical ORFs in line 382 but I can't see any such text. Can they please confirm the statement that has been included.

We apologize for this omission. We have now included references and an explicit paragraph at lines 403-406.

2. I don't see code in the supplied github repository for the TMHMM analysis as included in Supplemental data file 1 sheet 6 (and on the website to generate signal peptide free structures). The authors should clarify the availability of this code.

We apologize and have now included the scripts in our GitHub repository and reference them at lines 431-440.

3. On line 293, the authors have included the PDB reference to entry 3EIG but have not included a citation to the primary literature reference (Volpato, J.P., Yachnin, B.J., Blanchet, J., Guerrero, V., Poulin, L., Fossati, E., Berghuis, A.M., Pelletier, J.N. (2009) J Biol Chem 284: 20079-20089). This is exceedingly disrespectful to our structural colleagues - they are to thank for all the experimental data that underpins the success of AlphaFold and related methodologies! The relevant primary literature reference should be included here and any other time a PDB entry is referenced.

We sincerely apologize for this oversight. We have added references at line 313 for 3EIG and line 315 for 1RX2 as well as in the Supplementary Figure legends 3 for 7NXP, 4 for 4UZB, and 5 for 1RX2.

Reviewer #2 (Remarks on code availability):

I would have preferred if the authors would have also included the scripts used to perform the structural predictions, in addition to the analysis scripts. I can't see that readers could repeat their analysis on another family of viruses without writing such scripts/code themselves, which seems like a missed opportunity. The code supplied for post-prediction structure analysis seems appropriately documented from a brief review.

We used publicly available software repositories for structure predictions in the versions and with the settings stated in the Methods section and ran them locally. Since LocalColabFold and AlphaFold 2 were not developed by us and we did not fork them (*i.e.* changed any of the code) we are hesitant to provide a copy on our Github page. The commands we used for running the software are well-documented on the respective Github repositories and we now provide links to them in the manuscript. The used parameters are detailed in the Methods section, and we have added a statement about how we accessed the sequence information fasta files from UniProt. See the updated “software” section at lines 420-428.

Reviewer #3 (Remarks to the Author):

I thank the authors for considering my comments and amending their manuscript. Broadly, I remain enthusiastic about the work and its value to the community.

I am, however, a little disappointed that the authors did not do more to increase the visual accessibility of their work, I expect there were creative solutions to some of these problems. For example, scrutiny of the webpage screen shots in Figure 1 require zooming in, at which point they're revealed to be quite low resolution (not a good advert for the web resource). In Figure 2A, the clusters have no discernible labels, consequently, they are simply collections of colorful dots. In fact, each dot in this plot has a label, but these are invisible at standard view (*i.e.* it is unlikely the reader will know that they are there) and they only become legible at maximum zoom. This is before we consider the color-blind compatibility.

Ultimately, this comes down to how the authors want to communicate their work and not the quality of the scientific content of the manuscript. Therefore, if the authors are satisfied with this presentation I do not see this as a barrier to publication. However, if the work is accepted, I would recommend these issues are returned to at the copyediting stage.

We highly appreciate the support by reviewer 3. We have now included inserts in Figure 1 that show enlargements of the website for better readability. For Figure 2A, we decided to include a statement in the respective figure legend that details can be read upon zooming in and that an interactive version can be found at <https://www.herpesfolds.org/herpesclusters>. Concerning color-blind compatibility, the 9 colors are distinguishable for individuals with anomalous trichromacy, dichromacy, and blue cone monochromacy. However, we would indeed like to rely on professional expertise during copy-editing for all figures.

Reviewer #3 (Remarks on code availability):

I remain satisfied with the code and data availability, particularly as the authors now provide all data in a Zenodo repository.

REVIEWERS' COMMENTS

Reviewer #1 (Remarks to the Author):

Paper looks OK to publish